# Rethinking Graph Neural Networks from a Geometric Perspective of Node Features

**Feng Ji**[1*]**, Yanan Zhao**[1*]**, Kai Zhao**[1]**, Hanyang Meng**[2]**, Jielong Yang**[2]**, Wee Peng Tay**[1]
[1]School of Electrical and Electronic Engineering, Nanyang Technological University, Singapore
[2]School of Internet of Things Engineering, Jiangnan University, Wuxi, China

## Abstract

Many works on graph neural networks (GNNs) focus on graph topologies and analyze graph-related operations to enhance performance on tasks such as node classification. In this paper, we propose to understand GNNs based on a feature-centric approach. Our main idea is to treat the features of nodes from each label class as a whole, from which we can identify the centroid. The convex hull of these centroids forms a simplex called the feature centroid simplex, where a simplex is a high-dimensional generalization of a triangle. We borrow ideas from coarse geometry to analyze the geometric properties of the feature centroid simplex by comparing them with basic geometric models, such as regular simplexes and degenerate simplexes. Such a simplex provides a simple platform to understand graph-based feature aggregation, including phenomena such as heterophily, over-smoothing, and feature re-shuffling. Based on the theory, we also identify simple and useful tricks for the node classification task.

## 1 Introduction

Graph Neural Networks (GNNs) have emerged as important tools in managing graph-structured data, commonly found in application areas related to social networks, traffic systems, and biochemical structures (Gilmer et al., 2017; Ying et al., 2018; Chen et al., 2020; Li et al., 2021; Chen et al., 2022; Liu et al., 2022; Brody et al., 2022; Shen et al., 2023; Kang et al., 2023; Liang et al., 2024). The fundamental idea is to apply the technique of message passing or information propagation, formally called graph convolution (M. Defferrard, 2016; Shuman et al., 2013; Ortega et al., 2018). In this process, each node aggregates information from its neighboring nodes and potentially itself. Such a process can be repeated for multiple iterations, and the resulting aggregated information is expected to capture key features and hence represent each node better.

However, it is observed that such an approach does not universally work for any graph-structured data. Consider the node classification task, i.e., each node is associated with a label, and the labels of a test set of nodes are to be determined. In graph-structured data with the heterophilic property (Pei et al., 2020; Bo et al., 2021; Yan et al., 2022) (i.e., there are many edges connecting nodes from different classes), basic graph convolution (M. Defferrard, 2016; Kipf & Welling, 2017) is demonstrated to be ineffective, and significant modification is needed to get reasonable results (Pei et al., 2020). In many subsequent works, rewiring becomes an important tool for handling graph heterophily, most notably based on high-frequency graph spectral analysis. Another well-known phenomenon is oversmoothing (Oono & Suzuki, 2020; Yan et al., 2022), which renders training a deep graph convolution neural network extremely challenging. Oono & Suzuki (2020) studies the phenomenon by exploring the topological information of the underlying graphs inherent in the graph spectra. Techniques such as skip connections (Li et al., 2019; Chen et al., 2020) and diffusion with memory (Kang et al., 2024) have been proposed to address oversmoothing. In this paper, we study the node classification task in GNN models that involves node feature aggregation. We provide a unified theoretical framework to understand and analyze GNN phenomena, including graph heterophily and oversmoothing mentioned above, while new observations are also discussed.

---

[*]F. Ji and Y. Zhao have contributed equally. Corresponding to: J. Yang (jyang022@e.ntu.edu.sg), W. P. Tay (wptay@ntu.edu.sg).

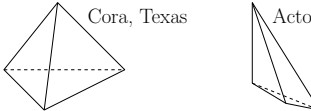

Figure 1: Illustrations of (parts of) the feature centroid simplexes associated with different datasets. Such a simplex is a graph-independent and model-agnostic intrinsic feature property. For example, we notice that simplexes of Cora, Texas datasets have a more regular shape. Though the Texas dataset is heterophilic, there are still GNN methods based on feature aggregation that achieves high accuracy on this dataset (improving GCN from $\approx 60\%$ to $\approx 90\%$). On the other hand, for the Actor graph, it does not seem that any approach can significantly improve upon GCN to the level we observe for the Texas dataset. We hope that our framework, based on studying the shape of the shown simplexes, provides a plausible explanation for this phenomenon.

We consider a graph as an extrinsic property of a dataset, and its construction can suffer from uncertainty and arbitrariness (Zhang et al., 2019; Dong & Kluger, 2023; Ji et al., 2023b). We regard node features as an intrinsic property, which is model-agnostic and does not change if the graph is modified. Unlike some works mentioned above that rely on graph structure and graph spectral properties, we focus mainly on graph-independent information that can be extracted from features, hoping to have some fundamental understanding of each dataset from a feature-centric perspective. In turn, we gain new insights into graph-based approaches.

Our main technical idea is to treat the features of nodes from each label class as a whole (in node classification), from which we can identify the centroid called the *class centroid*. We see, from explicit datasets, that the *unprocessed feature* of a node $v$ (with label $c$) needs not to be close in Euclidean distance to the class centroid of $c$. However, *aggregating features* of nodes belonging to the same class may output a feature much closer in distance to the class centroid. This prompts us to study each class centroid as the sole representative of its label class.

For this purpose, we take a geometric approach and consider the convex hull of these class centroids, forming a simplex called the *feature centroid simplex*, which is a high-dimensional generalization of a triangle (see Fig. 1). We borrow ideas from coarse geometry (Bridson & Häfliger, 2011) to analyze the geometric properties of the feature centroid simplex, by comparing them with basic geometric models, e.g., regular simplexes and degenerate simplexes. Such a geometric consideration provides a simple platform for us to understand graph-based feature aggregation and gain new insights into GNN phenomena. Moreover, different basic geometric models correspond to different intrinsic properties of features, from which we explore possible explanations for the "hardness" of datasets.

*In summary*, we propose to interpret the *role of the graph* as used to determine how aggregation is performed, which mitigates feature variance. Thus the *feature centroid simplex* is enough to capture essential intrinsic dataset properties. We use new geometric tools to study such a simplex. Though our work has limitations (Appendix G), we hope that it draws more attention to the study of "whether current GNN approaches have reached, to a certain extent, the *Bayesian error rate* for different datasets". Our main contributions are summarized as follows:

- We study GNN phenomena through the lens of node features instead of graph topologies. We introduce the notion of feature centroid simplex and develop a theoretical framework based on coarse geometry and probability theory. It provides us with tools for understanding and analyzing GNN models.

- We give alternative explanations to known observations and gain new insights into why it is hard to achieve significant results for certain datasets by any means. Moreover, we justify the effectiveness of some well-known GNN models from our perspective.

- Based on these theoretical findings, we identify surprisingly simple graph-independent tricks that are demonstrated to enhance the node classification performance of GNN models on certain datasets. We also explain why they work or fail on different datasets.

Due to space constraints, a notation list and proofs are delegated to Appendix A and B. We provide substantial numerical evidence to support our claims in Appendix E.

## 2 THE FEATURE SIMPLEX AND ITS COARSE MODELS

In this section, we present the probabilistic model for the feature space of each label class and propose a geometric framework to study its properties. As a preview, we use the topological concept of "simplex" to capture essential information about node features and introduce the notion of quasi isometry as the main tool for the study in the next section.

### 2.1 THE PROBABILISTIC FEATURE MODEL AND FEATURE SIMPLEX

Assume that $G = (V, E)$ is a connected, undirected graph of size $n = |V|$. We consider the following feature model. Each node $v_i$ is associated with an $m$-dimensional feature $x^i = (x_1^i, \ldots, x_m^i)^\mathsf{T}$ in $[0, 1]^m \subset \mathbb{R}^m$. Let $C$ be the finite set of class labels. We assume that node features $x^1, \ldots, x^n$ are identically independently generated via the following steps:

S1 For each (node) index $i = 1, \ldots, n$, a class $c$ is randomly chosen according to a prescribed distribution $\gamma$ (e.g., uniform distribution).

S2 A feature is generated according to a prescribed class-specific distribution $\gamma_c$.

To avoid technicalities, we provide an informal version of the geometric lemma to convey the key message. In Appendix B, we give a detailed specification of $\gamma_c$ and a formal statement of the lemma.

**Lemma 1** (Informal). *With high probability, $\{x^1, \ldots, x^n\}$ are in a* convex position, *i.e., none of $x^i$ is in the convex hull of the remaining feature vectors.*

We *assume* $\{x^1, \ldots, x^n\}$ are in a convex position for subsequent discussions. We have verified that the lemma holds for all datasets used in Section 5 below. Intuitively, the features are random points either on the vertices of a unit hypercube or in a high-dimensional unit ball. In either case, they are in a convex position with high probability due to the high-dimensional stochastic separation phenomenon (Gorban et al., 2018).

For each node $v_i$, recall $x^i$ is its feature vector, and let $c_i \in C$ be its label class.

**Definition 1.** *For each class $c$, denote $D_c = \{v_i \in V \mid c_i = c\}$ of size $n_c = |D_c|$ and define its feature convex hull as $\Delta_c = \mathrm{conv}(\{x^i \mid v_i \in D_c\})$, the convex hull of $\{x^i \mid v_i \in D_c\}$.*

By the assumption following Lemma 1, $D_c$ is the vertex set of the *convex polytope* $\Delta_c$. Moreover, $\Delta_c$ is a $(n_c - 1)$-*simplex* as it is homeomorphic to the *standard* $(n_c - 1)$-*simplex* (Hatcher, 2002, p. 9):

$$\Delta_{n_c} = \Big\{ (y_1, \ldots, y_{n_c})^\mathsf{T} \in [0, 1]^{n_c} \sum_{1 \leq i \leq n_c} y_i = 1 \Big\}. \tag{1}$$

For a simplex, a useful representative is its centroid.

**Definition 2.** *For each class $c$, let $e_c$ be the* probabilistic centroid *of the class defined as $e_c = \mathbb{E}_{x \sim \gamma_c}[x]$. Moreover, the* geometric centroid $g_c$ *of the samples of class $c$ is $\frac{1}{n_c} \sum_{v_i \in D_c} x^i \in \Delta_c$.*

Intuitively, the probabilistic and geometric centroid are close to each other if $n_c$ is large. More precisely, we have the following simple observation.

**Lemma 2.** *There is a constant $K$ independent of $n$ such that for $\delta > 0$ that is sufficiently small, $\mathbb{P}(\|e_c - g_c\| > \delta) \leq \exp(-K n_c \delta^2 + 1/4)$.*

*Convention*: This observation will be used subsequently, and we will refer to the same $K$ and $\delta$ in Lemma 2 without further mention.

Probabilistic centroids $\{e_c \mid c \in C\}$ are useful for theoretical analysis. However, in practice, we do not gain access to them. By Lemma 2, if we have sufficiently many samples for each class, we do not lose much information by using the geometric centroid $g_c$ as a proxy of $e_c$.

Recall that the objective of the node classification problem is to assign a discrete probability distribution $\mu_v = \{y_c \in [0, 1] \mid c \in C\}$ to each node $v \in V$. Geometrically, such a distribution $\mu_v$ belongs to the *probability simplex* $\Delta_C = \mathrm{conv}(\{p_c \in \mathbb{R}^{|C|} \mid c \in C\})$, which is the standard $(|C| - 1)$-simplex (here, $p_c$ is the one-hot vector of label class $c$). Ideally, features near $e_c \approx g_c \in \mathbb{R}^m$ (cf. Lemma 2) should be matched with $p_c$ via a learning model (illustrated in Fig. 2). This prompts us to study the following simplexes and analyze their relations with $\Delta_C$.

**Definition 3.** *The* feature centroid simplex *is the* $(|C| - 1)$*-simplexes* $\Delta_e = \text{conv}(\{e_c \,|\, c \in C\})$ *and its proxy is* $\Delta_g = \text{conv}(\{g_c \,|\, c \in C\})$.

In this next subsection, we discuss *coarse geometry*, which serves as the main tool to study feature centroid simplexes and GNNs.

## 2.2 COARSE GEOMETRY AND STANDARD MODELS

Recall that $\Delta_C$ has the ideal shape of being a *regular simplex* of equal side-length $\sqrt{2}$, both $\Delta_e$ and $\Delta_g$ are unlikely to be so. However, if they approximately have a regular shape, we can already draw useful conclusions (cf. Section 3). For this, we borrow ideas from coarse geometry (Bridson & Häfliger, 2011), from which we modify the notion of quasi-isometry (Bridson & Häfliger, 2011, p. 138) to obtain the following simplified version.

**Definition 4.** *Given* $\epsilon \geq 0$*, simplexes* $\Delta_y = \text{conv}(\{y^1, \ldots, y^k\})$ *and* $\Delta_z = \text{conv}(\{z^1, \ldots, z^k\})$ *are* $\epsilon$*-quasi isometric, denoted by* $\Delta_y \approx_\epsilon \Delta_z$*, if for any* $(a_1, \ldots, a_k)^\mathsf{T}, (b_1, \ldots, b_k)^\mathsf{T} \in \Delta_{k-1}$ *the standard* $(k-1)$*-simplex (see* (1)*), we have*

$$-\epsilon \leq \| \sum_{1 \leq i \leq k} a_i y^i - \sum_{1 \leq i \leq k} b_i y^i \| - \| \sum_{1 \leq i \leq k} a_i z^i - \sum_{1 \leq i \leq k} b_i z^i \| \leq \epsilon. \tag{2}$$

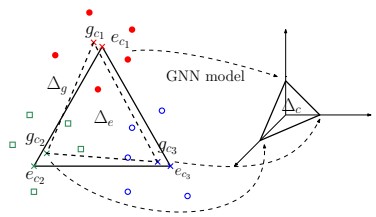

Figure 2: A schematic picture of the geometric concepts, e.g., centroids, $\Delta_g, \Delta_e, \Delta_C$. Nodes with different colors/shapes represent features belonging to different classes.

Being $\epsilon$-quasi isometric essentially means that the distance between any pair of points in $\Delta_y$ is close, up to an $\epsilon$ error, to the distance between the corresponding pair of points in $\Delta_z$. Hence, geometrically, $\Delta_y$ is a small perturbation of $\Delta_z$. While such a condition is usually hard to verify directly, we have the following version that only requires checking the vertices of the simplexes.

**Definition 5.** *A set of points* $\{y^1, \ldots, y^k\}$ *is* vertex $\epsilon$*-quasi isometric to* $\{z^1, \ldots, z^k\}$ *if for any* $i \neq j$*, we have* $\|(y^i - y^j) - (z^i - z^j)\| \leq \epsilon$.

**Lemma 3.** *Suppose* $\{y^1, \ldots, y^k\}$ *is vertex* $\epsilon$*-quasi isometric to* $\{z^1, \ldots, z^k\}$*. Then we have* $\Delta_y = \text{conv}(\{y^1, \ldots, y^k\}) \approx_\epsilon \Delta_z = \text{conv}(\{z^1, \ldots, z^k\})$.

Recall that two simplexes are congruent to each other if every pair of corresponding edges has the same length. Hence, we can use edge lengths of $\Delta_y, \Delta_z$ to probe their quasi-isometric relation. In Appendix E.2, we analyze the shapes of $\Delta_g$ for different datasets using the above notions.

**Model shapes** The basic idea of the paper is to compare feature centroid simplexes with standard geometric models. As a preview of the next section, $\Delta_y$ will be the subject we want to study, e.g., $\Delta_g$ and $\Delta_e$, while $\Delta_z$ will be a model simplex of special shapes. We end this section by describing some of these model shapes to be used later.

We have seen *regular simplexes*, which are simplexes with equal side lengths. A related notion is the following.

**Definition 6.** *Points* $\{x^0, x^1, \ldots, x^k\}$ *are said to form a* $(r_1, r_2)$*-regular* $k$ simplex *for* $r_1, r_2 > 0$ *if the base* $\Delta' = \text{conv}(\{x^1, \ldots, x^k\})$ *is a regular* $(k-1)$*-simplex of side length* $r_1$*, and* $\|x^0 - x^i\|$ *is the same for every* $1 \leq i \leq k$*. Moreover, the distance from* $x^0$ *to the centroid of* $\Delta'$ *is* $r_2$*. We call* $\text{conv}(\{x^0, x^1, \ldots, x^k\})$ *the associated* $(r_1, r_2)$*-regular* $k$*-simplex.*

For a preview, $x^0$ shall be used as a comparison point to the origin $0$.

There are also examples where $\Delta_e$ is "thin", i.e., some sides have much smaller lengths than others.

**Definition 7.** *We call a* $(k-1)$*-simplex* $\Delta$ *with vertices* $z^1, \ldots, z^k$ degenerate *if for some* $i \neq j$*,* $z^i = z^j$ *(see Appendix E.2 for a more general definition).*

We shall see such a shape corresponds to a "hard" dataset.

## 3 THE GEOMETRY OF GNN MODELS

In this section, we use the geometric setup of Section 2 to analyze the GNN convolution layer. Formally, a GCN layer consists of a convolution layer, a linear layer, and a ReLU activation. From empirical evidence in Appendix E Table 7, we observe that including the ReLU activation does not improve GCN performance in some datasets. Moreover, a 1-layer model is only slightly worse than a 2-layer model. Hence, to simplify the theoretical analysis in this section, we consider a 1-layer model without ReLU.

**Models of eventual GCN type**   Though our theoretical analysis focuses on the simplified GCN as above, the discussion in this section can be extended to more sophisticated models. For this, we define a model $\mathcal{M}$ to be of *eventual GCN type* if there exists a weighted graph $G'$ (possibly different from the original graph $G$) such that the output of $\mathcal{M}$ is the same as a GCN model applied to $G'$, with the exact same input features. For such a model, our framework can be used directly to analyze $\mathcal{M}$, in conjunction with the properties of $G'$. As a preview, let $N_v$ be the neighbors of a node $v$. If $G'$ is weighted, it suffices to replace $|N_v|$ in various expressions below with $1/\|W_v\|^2$, where $W_v = (w_{v_i,v})_{v_i \in N_v}$ is the edge weight vector and $w_{v_i,v}$ is the weight of the edge $(v_i, v)$ (using results in Barber (2024)).

Many GNN models are of (or related to) eventual GCN type. For example, attention-based models (Veličković et al., 2018; Lee et al., 2022; Lv et al., 2021) essentially generate a weighted graph for convolution, where the edge weights are the attention scores. There are models having rewiring or edge deletion as their components (Rong et al., 2020; Suresh et al., 2021; Topping et al., 2022; Ji et al., 2023a). They are of eventual GCN type, where $G'$ is the new graph topology. In a neural diffusion model (Chamberlain et al., 2021; Rusch et al., 2022; Zhao et al., 2023), the basic idea is to iteratively use a forward Euler feature updating scheme to model the solution of a diffusion equation. The update rule resembles a GCN convolution on a weighted graph that may change in each iteration. The adaptive nature adds flexibility that may mitigate unfavorable phenomena such as oversmoothing. These models can also be analyzed using our framework analogous to our discussion above by keeping track of the shape of $\Delta_e, \Delta_g$ in updates.

### 3.1 FEATURE AGGREGATION

Recall that for each node $v$, let $N_v$ be the 1-hop neighbor of $v$, including $v$ itself. The convolution (M. Defferrard, 2016) updates the feature of $v$ as

$$y_v = \frac{1}{|N_v|} \sum_{v_i \in N_v} x^i. \tag{3}$$

We are interested in the location of $y_v$ in the feature space $\mathbb{R}^m$, particularly its proximity to the vertices and special points on the simplex $\Delta_e$. To state the following main result, for $c \in C$, let $N_{v,c} = N_v \cap D_c$ and define its *mixed centroid*

$$e_{N_v} = \frac{1}{|N_v|} \sum_{v_i \in N_v} e_{c_i}. \tag{4}$$

**Theorem 1.**     *(a) We have $\mathbb{P}(\|y_v - e_{N_v}\| \leq |C|\delta) \geq \left(1 - \exp(-K\delta|N_v| + 1/4)\right)^{|C|}$.*

*(b) Suppose there is $\epsilon \geq 0$ and a regular $(|C| - 1)$-simplex $\Delta = \mathrm{conv}(\{u_c \mid c \in C\})$ with (the common) side length $r$ such that: $\{e_c \mid c \in C\}$ is vertex $\epsilon$-quasi isometric to $\{u_c \mid c \in C\}$. Then, for any $c \in C$, we have*

$$\mathbb{P}\left(\|y_v - e_c\| \geq \frac{r}{\sqrt{2}} d_{v,c} - \epsilon - |C|\delta\right) \geq \left(1 - \exp(-K\delta|N_v| + 1/4)\right)^{|C|},$$

*where $d_{v,c}^2 = (1 - \frac{|N_{v,c}|}{|N_v|})^2 + \sum_{c' \neq c} (\frac{|N_{v,c'}|}{|N_v|})^2$.*

**Discussions**   Notice that the probability bound in Theorem 1 (a) only involves $|N_v|$. Intuitively, this means that if a node has sufficiently many neighbors (disregarding their label classes), then with high

probability, its aggregated feature is close to the point in $\Delta_e$ aggregated, in the same way, from its vertices. In (b), the model simplex, which is a regular one, plays an essential role. Notice that $d_{v,c} \approx 0$ if and only if $|N_{v,c}| \approx |N_v|$. Otherwise by (b), if $r \gg \epsilon$, then with high probability $\|y_v - e_c\|$ has a large lower bound. As a consequence, it may be prudent during training to avoid directly fitting a node, with many neighbors from different classes, to its ground-truth class. We propose tricks based on this and demonstrate that they work better for datasets with more regular $\Delta_g$ in Section 5 (see also discussions in Section 4 and a thorough analysis of simplex shapes in Appendix E.2).

## 3.2 THE LEARNED LABEL DISTRIBUTION

Following feature aggregation, the convolution layer is a linear transformation. Recall that the goal of any model is to generate an element in the probability simplex $\Delta_C$ for each node $v \in V$. We want to investigate the probability distribution a linear layer following the feature aggregation can possibly generate.[1] For this, we first introduce a less strict type of simplexes than regular simplexes.

We have used a regular simplex to approximate the simplex $\Delta_e$. As illustrated in Appendix E.2 Fig. 7, for many datasets, centroids in $\{g_c \,|\, c \in C\}$ are approximately equal in distance to 0. Hence, $(r_1, r_2)$-*regular simplexes* introduced in Definition 6 are reasonable model shapes. For the next result, we introduce the following quantity for any $(r_1, r_2)$-regular $k$-simplex.

Suppose $\{x^0, x^1, \ldots, x^k\}$ form an $(r_1, r_2)$-regular $k$-simplex $\Delta$ with base $\Delta'$. We define:

$$\rho(r_1, r_2) = \min\left\{ \frac{r_1}{\sqrt{2}(k!)^{1/(k-1)}}, r_2^{1/(k-1)} k^{1/2(k-1)} \right\}, \tag{5}$$

which is independent of the locations of $x^0, x^1, \ldots, x^k$.

Recall features $x^1, \ldots, x^n$, and vectors $y_v, e_{N_v}$ defined earlier in (3), (4). We have the following.

**Theorem 2.** *Recall $\Delta_C = \mathrm{conv}(\{p_c \,|\, c \in C\})$. Assume that $\{0\} \cup \{e_c \,|\, c \in C\}$ is vertex $\epsilon$-quasi isometric to $\{0\} \cup \{z_c \,|\, c \in C\}$ that form an $(r_1, r_2)$-regular simplex. Suppose $\sqrt{|C|}\epsilon < \rho(r_1, r_2)$ and define $\rho(r_1, r_2, \epsilon) = (\rho(r_1, r_2) - \sqrt{|C|}\epsilon)^{-1} > 0$. Then there is a linear transformation $L$ from the feature space to $\mathbb{R}^{|C|}$ such that the following holds.*

*(a) For each $c \in C$, if $\widehat{y}_c = \arg\min_{y \in \mathbb{R}^{|C|}} \sum_{v_i:\, \text{class } c} \|L(x^i) - y\|$, then*

$$\mathbb{P}\Big( \|p_c - \widehat{y}_c\| \leq \rho(r_1, r_2, \epsilon)\delta \Big) \geq 1 - \exp(-K n_c \delta^2 + 1/4).$$

*(b) For any node $v$, let $\mu_v \in \Delta_C$ be the neighborhood class label probability distribution of $v$, i.e., $\mu_v(c) = |N_{v,c}|/|N_v|$. Then*

$$\mathbb{P}\Big( \|L(y_v) - \mu_v\| \leq \rho(r_1, r_2, \epsilon)|C|\delta \Big) \geq (1 - \exp(-K\delta|N_v| + 1/4))^{|C|}.$$

**Discussions** We point out that $L(y_v)$ in Theorem 2 (b) corresponds exactly to one GCN layer. Intuitively, the result essentially claims that for each node $v$, the linear fitting $L$ and hence the GCN layer generates *a probability distribution that is almost the same as the neighborhood class label distribution of $v$*, with high probability. A numerical verification using a thought experiment is in Appendix E.3, showing if $\nu_v$ is known, then forcing the output distribution to be similar to $\nu_v$ solves the classification problem.

Notice by (5), the factor $\rho(r_1, r_2, \epsilon)$ in the upper bound is small if $r_1, r_2 \gg \epsilon$. This occurs again if $\{0\} \cup \{e_c \,|\, c \in C\}$ forms a simplex of relatively regular shape. Therefore, in GNN model design, re-shaping $\Delta_e$ into a more regular shape is a strategy worth considering (see Section 5). In the next subsection, we discuss the consequence if $\Delta_e$ does not have a regular shape.

## 3.3 DEGENERATE MODEL SIMPLEXES

We have thus far discussed only feature centroid simplexes having an almost regular shape. In this subsection, we consider the contrary of having an almost degenerate shape. For the next result, recall $y_v, e_{N_v}$ defined in (3) and (4).

---

[1]This simplified setting is convenient for theoretical analysis. In a GNN model, softmax is usually used and we discuss its effect Appendix H.

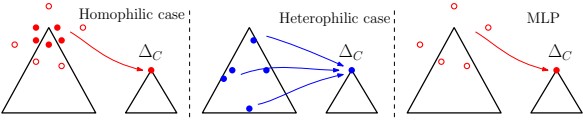

Figure 3: In the homophilic case, the aggregated features (solid disc, while circles are the original features and they are more scattered) are matched approximately to the correct vertex in the probability simplex, while in the heterophilic case, the features are no longer suitable to fit with vertices of the probability simplex. Therefore, in the heterophilic case, GCN may even underperform against a vanilla MLP, though for the latter the features are more scattered.

**Corollary 1.** *Suppose there is an $\epsilon \geq 0$ and a degenerate simplex $\Delta = \mathrm{conv}(\{u_c \,|\, c \in C\})$ (with $u_{c_i} = u_{c_j}, i \neq j$) such that: $\{e_c \,|\, c \in C\}$ is vertex $\epsilon$-quasi isometric to $\mathrm{conv}(\{u_c \,|\, c \in C\})$. Define $\overline{e_{N_v}} \in \Delta_e$ by replacing $e_{c_i}$ with $e_{c_j}$ and vice versa. Then*

$$\mathbb{P}\Big(\|y_v - \overline{e_{N_v}}\| \leq |C|\delta + \epsilon\Big) \geq \big(1 - \exp(-K\delta|N_v| + 1/4)\big)^{|C|}.$$

**Discussions**   The result suggests that if $\Delta_e$ is almost a degenerate simplex, i.e., it is thin, then there is no way to very accurately identify the summands in $y_v$ by observing it. Notice that for such a $\Delta_e$, if it has sides whose length is much larger than $\epsilon$, then it cannot be reasonably modeled by a regular simplex as in Theorem 1 and Theorem 2. The quasi isometric model of $\Delta_e$ is an intrinsic property of features independent of the graph. Therefore, many models of eventual GCN type, which rely only on altering the aggregation mechanism, cannot resolve this inherent issue of features. This is verified by the feature re-shuffling phenomenon in the next section. Datasets (e.g., the Actor dataset) that satisfy the conditions of Corollary 1 are shown in Appendix E.2 and studied further in Appendix E.4, Appendix E.6. We explain the general poor performance of GNN models on these datasets.

## 4   PHENOMENA IN GNNS

We first summarize general insights derived from the feature-centric perspective in Section 3 to guide our discussions: (a) Feature aggregation is essential to reduce feature variance (see a discussion on the use of normalized adjacency matrix in Appendix H). (b) The efficiency of feature aggregation depends on the shape of the feature centroid simplex, which is an intrinsic graph-independent property. (c) We should aim to aggregate features of nodes of the same class in both training and testing, which, *a priori*, is the role of the graph structure. (d) However, (c) is not always easily achievable. In this case, the model output label distribution should be interpreted in conjunction with the aggregation mechanism and the graph structure.

**Homophilic graphs v.s. heterophilic graphs**   If a graph is homophilic (illustrated in Fig. 3), then for most of the nodes, their neighbors belong to the same type, i.e., for a node $v$ with label $c$, $|N_{v,c}|/|N_v| \approx 1$ and $|N_{v,c'}|/|N_v| \approx 0$ for any $c' \neq c$. In this case, $e_{N_v} \approx e_c$ and by Theorem 1 (a), the aggregated feature $y_v$ is close to $e_c$ with high probability. Hence, $y_v$ can be fitted close to $p_c$ via the linear map $L$ as in Theorem 2, while the output probability distribution of a node $v$ is close to the neighborhood class label distribution $\mu_v$. Test error will thus likely occur for those nodes whose neighbors contain a significant number of nodes from different classes. For a simple verification (on Cora), for each test node $v_i$ of class $c_i$, we compute $\mu_{v_i}(c_i)$ (defined in Theorem 2(b)). Its average value is $\approx 0.911$ for correctly classified nodes, while only $\approx 0.668$ for misclassified nodes.

On the other hand, in a heterophilic graph (illustrated in Fig. 3), many nodes have neighbors belonging to different label classes, i.e., for a node $v$ of label class $c$, $|N_{v,c}|$ is significantly smaller than $N_v$. In this case, $d_{v,c}$ in Theorem 1 is significantly larger than 0, which implies that the aggregated feature $y_v$ is bounded away from $e_c$ with high probability by Theorem 1 (b). It thus causes significant error if we try to fit $y_c$ with the vertex $p_c$ in the probability simplex $\Delta_C$. This is why simple MLP sometimes works better as shown in Yan et al. (2022). In Section 5, we focus on heterophilic graphs to address the above issues.

**Oversmoothing**   It is observed in Oono & Suzuki (2020) that oversmoothing is due mainly to the Markovian property of feature aggregation. Oono & Suzuki (2020) explains the phenomenon from a

graph spectral perspective. Here, we use our geometric perspective of node features to provide another point of view, which can be easily visualized. If we disregard the ReLU function, for a $T$-layer model, we essentially consider $T$-steps of neighborhood aggregation, followed by a linear transformation. We inspect the change of the simplex $\Delta_g = \mathrm{conv}(\{g_c \mid c \in C\})$. Recall that $g_c = \frac{1}{n_c} \sum_{v_i \in D_c} x^i \approx e_c$ by Lemma 2. After the aggregation step, $g_c$ becomes

$$g_c^{(1)} = \frac{1}{n_c} \sum_{v_i \in D_c} \frac{1}{|N_{v_i}|} \sum_{v_j \in N_{v_i}} x^j = \frac{1}{n_c} \sum_{v_i \in D_c} \frac{1}{|N_{v_i}|} \sum_{c' \in C} \sum_{v_j \in N_{v_i,c'}} x^j$$

$$= \sum_{c' \in C} \Big( \frac{1}{n_c} \sum_{v_i \in D_c} \frac{1}{|N_{v_i}|} \sum_{v_j \in N_{v_i,c'}} x^j \Big).$$

Let $\alpha_{c,c'} = \frac{1}{n_c} \sum_{v_i \in D_c} |N_{v_i,c'}|/|N_{v_i}|$. For each $c$, there are some $c' \neq c$ such that $\alpha_{c,c'} \neq 0$ as each neighbor of $D_c$ belongs to a different class. Hence, $\alpha_{c,c'} < 1$. Then $g_c^{(1)}$ from above becomes $\sum_{c' \in C} \alpha_{c,c'} \sum_{v_j \in D_{c'}} \Big( \frac{1}{\alpha_{c,c'} n_c} \sum_{v_i \in D_c, v_j \in N_{v_i}} \frac{1}{|N_{v_i}|} \Big) x^j$. By a mean-field type assumption, we assume for $v_j$ belonging to the same class, the non-zero coefficients in front of $x^j$ are approximately the same. The coefficient is non-zero if $v_i$ is within the 1-hop neighbor of the entire set $D_c$. If there are many such $v_j$'s, then $\sum_{v_j \in D_{c'}} \big( \frac{1}{\alpha_{c,c'} n_c} \sum_{v_i \in D_c, v_j \in N_{v_i}} \frac{1}{|N_{v_i}|} \big) x^j \approx e_{c'}$. Therefore, *in conclusion*, $g_c^{(1)} \approx e_c^{(1)} = \sum_{c' \in C} \alpha_{c,c'} e_{c'}$, i.e., after a single layer of aggregation, the average $g_c^{(1)}$ of the updated feature $\{y_v, v \in D_c\}$ is approximately $e_c^{(1)}$. The update coefficients $\{\alpha_{c,c'}, c' \in C\}$ depend only on the dataset but not on the iteration number. By the same argument, define iteratively $e_c^{(t)} = \sum_{c' \in C} \alpha_{c,c'} e_{c'}^{(t-1)}$. We study how $\Delta_e^{(t)} = \mathrm{conv}(\{e_c^{(t)}, c \in C\})$ evolves for $t \geq 1$ to understand the features updates in each iteration. The above formula implies that the vertices of $\Delta_e^{(t)}$ are updated via the matrix $M = (\alpha_{c,c'})_{c,c' \in C}$.

**Lemma 4.** *$M$ is a stochastic matrix and it has a unique stationary distribution*

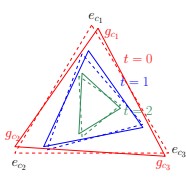

Figure 4: An illustration of oversmoothing. The simplex $\Delta_e^{(t)}$ in each iteration is represented by a (colored) triangle. After enough iterations, it shrinks to the center.

As a consequence of the lemma, if $(s_c)_{c \in C}$ is the stationary distribution of $M$, then each $e_c^{(t)}$ converges to $\sum_{c \in C} s_c e_c$. Therefore, we have the oversmoothing phenomenon as $\Delta_e^{(t)}$ converges to a single point at $t \to \infty$. A schematic picture is shown in Fig. 4.

We numerically verify by training a deep GCN and study the volume $\mathrm{Vol}(\Delta_g^{(t)})$ of the resulting $\Delta_g^{(t)}$ after the $t$-th layer, expecting to see it diminishing. For Cora, after 4 layers, the (average) volume is reduced to $\approx 1.7\%$ of the initial (average) volume. For Texas, after 7 layers, the (average) volume is reduced to $\approx 2.6\%$ of the initial (average) volume. The observations support the theory.

**Feature re-shuffling** Our result in Section 3.3 predicts the following: for a dataset, if the feature centroid simplex $\Delta_e$ is almost degenerate, then it imposes a significant challenge for the classification task. This phenomenon is observed for the Chameleon, Squirrel, and Actor datasets (see results in Section 5 and also Appendix E for further discussions). By the definition of degeneracy, there are at least two classes $c_i, c_j$, whose associate centroids $e_{c_i}, e_{c_j}$ are very close. Therefore, it is expected that if we re-shuffle the features of nodes of classes $c_i, c_j$ (possibly resulting in nodes of class $c_i$ having features of class $c_j$), there will be no significant drop or boost in the classification result. This is verified by the results in Table 1. The study reveals the model-agnostic intrinsic difficulties for the classification task from the feature-centric geometric perspective.

For easier datasets whose $\Delta_e$ is more regular (e.g., Cora, Citeseer), re-shuffling nodes belonging to the *same class* may even improve model performance. A graph is usually constructed with resulting edges connecting nodes with similar features. However, aggregating similar features is less useful in reducing feature variance. Therefore, re-shuffling may overcome such a shortcoming and after re-shuffling, the aggregated features can be closer to the centroid (see Fig. 5). This is verified on Cora and Citeseer in Table 1, and similarly observed in Lee et al. (2024).

Table 1: Based on the study in Appendix E.4, we re-shuffle features of class $(c_1, c_2)$, $(c_1, c_3)$ and $(c_2, c_3)$ using vanilla GAT, GAT, ACM-GCN (as top performers in Table 2 below) for the Chameleon, Squirrel and Actor datasets respectively. Re-shuffling does not harm the performance. For Cora and Citeseer, GCN is applied, and re-shuffling significantly improves the performance.

|  | Chameleon | Squirrel | Actor | Cora | Citeseer |
|---|---|---|---|---|---|
| Pre re-shuffle | 71.64±2.69 | 61.67±5.00 | 34.39±1.31 | 80.95±0.41 | 70.91±0.48 |
| Post re-shuffle | 71.27±1.93 | 62.74±3.88 | 34.12±1.35 | 86.36±0.26 | 79.36±0.39 |

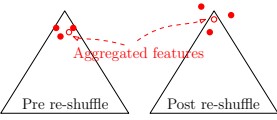

Pre re-shuffle    Post re-shuffle

Figure 5: Re-shuffling makes features of nodes connected by an edge less similar. The resulting aggregated feature is thus closer to the class centroid.

*In conclusion*, the study supports the presumption that a key role of the graph is for feature aggregation to reduce feature variance, and the aggregation alone does not resolve intrinsic issues arising from the suboptimal shape of the simplex $\Delta_e$.

## 5 EXPERIMENTS

**Simple tricks**  The theoretical findings suggest simple generic tricks might be employed to enhance the performance of GNN models. We propose two such *graph independent* tricks with justification:

(1) Recall by Theorem 1 (a), the aggregated feature of a training node $v_i$ is approximate $e_{N_{v_i}}$ defined in (4). For $e_{N_{v_i}}$ to be close to $e_{c_i}$, it is desirable to have more nodes in the neighbor of $v_i$ to have the same label $c_i$. A simple trick for this is to independently randomly *add edges* between *training nodes* of the same class, with probability $\eta$ (a hyperparameter).

(2) By Theorem 1 (b) and discussions in Section 4, the aggregated feature $y_{v_i}$ of a training node $v_i$ can deviate from $e_{c_i}$ if it has a large percentage of neighbors not from the same class $c_i$ (see also Fig. 3). Thus, the better $y_{v_i}$ fits with $p_{c_i}$, the more error it causes in testing. We thus propose a *very early stopping* by training $\mathcal{E}$ (a hyperparameter) epochs for a small $\mathcal{E}$.

The proposed tricks fall into a few common generic categories of machine learning techniques. In Appendix C: Related Works, we discuss how they differ from existing methods, belonging to the same categories, at the conceptual level.

The tricks are model-agnostic and can be applied (either separately or jointly) to most GNN models. Given a base model $\mathcal{M}$, we use "$\mathcal{M}$-AE" to denote its modification when the tricks are incorporated. The tricks are simple and have minimal effect on the computation complexity. We consider GCN (Kipf & Welling, 2017), GAT (Veličković et al., 2018), ACM-GCN (Luan et al., 2022), GraphCON (Rusch et al., 2022), CDE (Zhao et al., 2023), and GloGNN (Li et al., 2022) as base models. Experimental details are in Appendix D. We *focus on heterophilic datasets*, which have rich structures to analyze (see results for *homophilic* and *large scale non-homophilous* datasets in Appendix F.1, which also contains discussions on the *feature scarce scenario*).

The results are shown in Table 2. Among 48 comparisons, the above tricks significantly (in terms of $p$-value) improve the performance in 34 instances, and the improvement is insignificant in 14 instances, mainly for Chameleon, Squirrel, and Actor datasets. As explained in Section 3.3, due to the feature centroid simplex being approximately degenerate, it does not seem that any model is particularly effective for these datasets (though some are relatively better).

In Appendix E.2, Appendix E.4, and Appendix E.5, we thoroughly analyze Chameleon, Squirrel, and Actor datasets. We explain why these are the more difficult datasets from our point of view. We also observe that it is more likely to mistakenly predict the class $c_j$ for the ground-truth class $c_i$ if $\|g_{c_i} - g_{c_j}\|$ is small, which is consistent with theory and the feature re-shuffling study in Table 1.

**The feature normalization trick**  In Appendix E Fig. 7 and Fig. 9, we observe that a common issue with Chameleon, Squirrel, and Actor datasets is that features of most nodes are not expressive in the sense that they have a small norm. The resulting simplex $\Delta_g$ thus has a less regular shape (cf. Section 3.3). We apply a simple normalization trick that normalizes each feature vector to have norm

Table 2: Node classification results(%). The best result for each dataset is highlighted in blue. "-I" and "-II" stand for 1 and 2 layers respectively.

| Method | Texas | Cornell | Wisconsin | Chameleon | Squirrel | Actor |
|---|---|---|---|---|---|---|
| GCN-I | 68.92±7.95 | 62.97±4.36 | 62.97±4.36 | 59.19±2.33 | 42.96±1.46 | 30.68±0.45 |
| GCN-I-AE | 79.45±8.30 | 69.73±7.63 | 71.37±7.66 | 63.54±4.16 | 45.06±6.22 | 34.30±2.89 |
| GCN-II | 58.11±7.07 | 60.54±9.30 | 56.47±7.98 | 65.42±2.02 | 48.51±1.82 | 27.39±1.22 |
| GCN-II-AE | 63.24±4.22 | 61.08±5.30 | 61.96±6.46 | 66.25±4.15 | 50.96±3.17 | 30.85±1.59 |
| GAT-I | 77.57±6.95 | 70.81±8.87 | 69.61±6.46 | 66.71±3.10 | 54.21±3.21 | 28.66±1.21 |
| GAT-I-AE | 81.08±6.51 | 74.05±10.69 | 71.96±8.59 | 68.93±5.98 | 55.02±5.55 | 32.35±1.87 |
| GAT-II | 65.14±10.50 | 60.81±10.89 | 61.57±7.03 | 71.64±2.69 | 61.67±5.00 | 28.09±1.60 |
| GAT-II-AE | 66.48±10.48 | 67.30±8.15 | 63.31±8.36 | 72.24±3.21 | 62.37±5.54 | 31.82±1.58 |
| ACM-GCN | 88.92±4.26 | 86.76±7.49 | 88.24±5.11 | 71.29±8.06 | 55.07±8.96 | 34.39±1.31 |
| ACM-GCN-AE | 90.81±5.30 | 88.91±6.56 | 91.18±6.46 | 73.38±9.78 | 58.08±9.65 | 42.60±3.93 |
| GraphCON | 88.65±4.65 | 82.97±4.37 | 87.84±4.54 | 64.30±3.45 | 43.34±4.00 | 30.26±1.53 |
| GraphCON-AE | 91.08±4.37 | 84.86±4.86 | 89.22±2.52 | 65.13±3.77 | 43.57±4.69 | 32.46±2.64 |
| CDE | 84.86±5.69 | 72.97±4.68 | 82.16±6.10 | 59.93±0.97 | 43.49±1.84 | 36.41±1.08 |
| CDE-AE | 85.41±5.01 | 75.68±5.27 | 85.69±3.83 | 60.09±2.27 | 44.01±1.69 | 36.58±1.05 |
| GloGNN | 82.16±5.57 | 82.97±4.02 | 84.90±5.55 | 69.78±2.38 | 57.61±1.28 | 37.11±1.57 |
| GloGNN-AE | 83.78±4.19 | 83.24±4.95 | 86.27±3.82 | 69.32±2.52 | 57.65±1.44 | 37.33±1.43 |

1. Intuitively, the procedure relatively increases the size of each component of features with a small norm. In turn, the shape of the simplex $\Delta_g$ may become more regular, which is supported by the analysis in Appendix E.6. We see its usefulness from Table 3. The trick is also shown to be very effective for the Ogbn-arxiv dataset in Appendix F.1 Table 10.

Table 3: Normalization (suffix: "-AEN") applied to $\mathcal{M}$-AE: the overall best model ACM-GCN.

| Method | ACM-GCN | ACM-GCN-AE | ACM-GCN-AEN | % ↑ over $\mathcal{M}$ |
|---|---|---|---|---|
| Chameleon | 72.29±8.06 | 73.38±9.79 | 75.22±9.99 | 4.05% |
| Squirrel | 55.07±8.96 | 58.08±9.65 | 59.47±11.41 | 7.99% |
| Actor | 34.28±2.76 | 42.60±3.93 | 47.62±6.38 | 38.9% |

**The output probability distribution**   In Theorem 2 (b), it is predicted that the output label distribution $\nu_v$ of a node $v$ is approximately its neighborhood label distribution $\mu_v$. We measure their difference by the mean of $\ell_v = \|\nu_v - \mu_v\|_1$ (the choice of $\|\cdot\|_1$ is explained in Appendix H), averaged over all test nodes. From the results in Table 4 (on GCN-I), we see that the tricks indeed improve the average $\ell_v$. As eventual GCN models change the graph structure, more results in Appendix F.3 show that most other models underperform in this metric.

As we show in Appendix E.3 and Fig. 10, knowing $\{\mu_v \mid v \in V\}$ accurately is sufficient for the node classification problem. In conjugation with Theorem 2 (b), we propose estimating $\{\mu_v \mid v \in V\}$ as an alternative objective for future work. It is possible to further reduce $\ell_v$ by *conformal prediction*, as a mean of distribution denoising. Details can be found in Appendix F.2.

Table 4: The performance of the tricks in the metric of average $\ell_v = \|\nu_v - \mu_v\|_1$.

| Method | Texas | Cornell | Wisconsin | Chameleon | Squirrel | Actor |
|---|---|---|---|---|---|---|
| GCN-I | 0.6874 | 0.7286 | 0.7204 | 0.7360 | 0.6240 | 0.9044 |
| GCN-I-AE | 0.6181 | 0.6482 | 0.5634 | 0.6934 | 0.5834 | 0.8877 |

# 6   CONCLUSIONS

In this paper, we propose a theoretical framework for understanding GNN models based on coarse geometric models of feature centroid simplex. It allows us to have a glimpse into theoretical explanations of GNN phenomena such as oversmoothing and feature re-shuffling, and to enhance our understanding of the mechanisms and fundamental limitations of GNN models. We discuss the limitations of our framework and propose future directions in Appendix G.

ACKNOWLEDGEMENT

This research is supported by the National Research Foundation, Singapore and Infocomm Media Development Authority under its Future Communications Research and Development Programme.

H. Meng and J. Yang are supported in part by the National Science Foundation of China (No.62106082).

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

## A    LIST OF NOTATIONS

For easy reference, we list the most used notations in Table 5.

Table 5: List of notations

| | |
|---|---|
| Graph, Vertex set, Edge set | $G$ (with subscripts), $V, E$ |
| Features and vectors | $x^i, y^i, z^i$ |
| Nodes | $v, v'$ |
| A set of coordinates | $\mathcal{J}$ (with subscripts) |
| A set of label classes | $C$ |
| Label classes | $c$ (with subscripts) |
| The set of nodes belong to a class $c$ | $D_c$ |
| Size of $D_c$ | $n_c$ |
| Probabilistic centroid (of a class $c$) | $e_c$ |
| Geometric centroid (of a class $c$) | $g_c$ |
| Convex hull | $\mathrm{conv}(\cdot)$ |
| Simplex | $\Delta$ (with subscripts) |
| The probability simplex | $\Delta_C$ |
| The feature centroid simplex and its proxy | $\Delta_e, \Delta_g$ |
| Aggregated feature | $y_v$ |
| 1-hop neighbors, neighbors of class $c$ | $N_v, N_{v,c}$ |
| Mixed centroid | $e_{N_v}$ |
| Neighborhood class label probability distribution | $\mu_v$ |

## B    THEORETICAL DISCUSSIONS AND PROOFS

To make Lemma 1 precise, we consider the following specific models for each $\gamma_c$.

For each class $c \in C$, we assume that there is a set of class-specific coordinates $\mathcal{J}_c$ of $\mathbb{R}^m$ such that the following holds:

(a) Let $S_c$ be the subset of $\mathbb{R}^m$ consisting of vectors $(x_1, \dots, x_m)^\mathsf{T}$ such that $x_j = 0$ if $j \notin \mathcal{J}_c$ and $x_j \in (0,1]$ if $j \in \mathcal{J}_c$. There is a probability distribution $\gamma_c$ on $S_c$ such that features of class $c$ nodes are drawn independently from $S_c$ according to $\gamma_c$.

(b) There is an $m_1$ such that if $c_1 \neq c_2$, then $|\mathcal{J}_{c_1} \cap \mathcal{J}_{c_2}| \leq m_1$.

(c) There is an $m_2 > m_1$ such that for each $c$, $|\mathcal{J}_c| \geq m_2$.

The range $[0,1]$ is adopted without loss of much generality since a component of a feature often represents a probability or a correlation. Most applications also have bounded feature component values, which can be normalized to be within $[0,1]$. For many datasets (e.g., those in Appendix D.1), the components of each initial feature vector are always either $0$ or $1$.

Model 1    For each feature $x$ belonging to the class $c$ and each axis index $i \in \mathcal{J}_c$, the $i$-th component $x_i$ of $x$ follows the Bernoulli distribution with parameter $p_{c,i}$, independently for different $i$.

Model 2    For each $i \in \mathcal{J}_c$, $x_i$ follows a continuous distribution on $[0,1]$ with density bounded by $b$, which is independent of $c$. For different $i$, the continuous distribution are independent but possibly non-identical.

For example, the first model is used for features in datasets such as Cora, Citeseer, Pubmed, while the second model is used for word embedding features and pixel values for CV. We can now state and prove the precise version of Lemma 1 as follows.

**Lemma 5.** *There is a $B < 1$ that depends on $b$ while is independent of $m, n$ such that the following holds: the probability that $F = \{x^1, \ldots, x^n\}$ are not in convex position is at most $nB^{m_2}/\sqrt{\pi}$.*

Unless $n$ is much larger than $m_2$, the features are in a convex position with high probability.

*Proof.* For Model 1, each feature is a vertex of the $m$-dimensional hypercube $[0,1]^m$, which is a convex set. Therefore, $F$ is always in a convex position.

For Model 2, suppose $x^j$ is in the convex combination of $F' \subset F$. Then we may assume that $x^j$ and $F'$ belong to the same class $c$, for otherwise, $x^j$ is the convex combination of the form $x^j = \sum_{x \in F'} a_x x, a_x > 0$. For some $x$, $x^j$ and $x$ belong to different classes. However, this is impossible as follows. Recall that each component of either $x^j$ or $x \in F$ is non-negative. Since $m_1 < m_2$, there is an index $i$ such that either $x_i^j$ or $x_i$ is zero, but not both. This is a contradiction.

Therefore, we have at most $n$ i.i.d. vectors belonging to an $m_2$ dimensional subspace of $\mathbb{R}^m$. According to the model for $\gamma_c$, such a probability is at most $nB^{m_2}$ (for some $B < 1$ independent of $m, n$) by Gorban et al. (2018) Example 4 and Theorem 2. This proves the lemma. $\qquad\square$

Lemma 2 follows essentially from concentration type of inequalities, while Lemma 6 below is due to the additional assumption on the feature model.

*Proof of Lemma 2.* We form independent random vectors $y^i = x^i - e_c$. As each component of $y^i$ is in $[-1, 1]$, $\mathbb{E}(\|y^i\|^2) = \sigma^2 < \infty$. The result follows now from the vector Bernstein inequality (Kohler & Lucchi, 2017, Lemma 18) applied to $\{y^i\}$ with $K = 1/8\sigma^2$. $\qquad\square$

For different classes $c, c'$, a lower bound on $\|e_c - e_{c'}\|$ tells us how different classes are separated away from each other. We have the following observation regarding this.

**Lemma 6.** *Assume that there is a universal lower bound $b'$ on the mean for each non-zero component (in $\mathcal{J}_c$) of all classes. Then $\|e_c - e_{c'}\| \geq (2m_2 - 2m_1)^{1/2}b'$. Moreover, we have $\mathbb{P}\big(\|g_c - g_{c'}\| \leq (2m_2 - 2m_1)^{1/2}b' - 2\delta\big) \leq \exp(-K(n_c + n'_c)\delta^2 + 1/2)$.*

*Proof.* By our assumptions on $\mathcal{J}_c$ and $\mathcal{J}_{c'}$, for vectors $e_c - e_{c'}$, there are at least $2m_2 - 2m_1$ non-zero entries. The absolute value of each such entry is at least $b'$. Therefore, $\|e_c - e_{c'}\| \geq (2m_2 - 2m_1)^{1/2}b'$. The second half follows from Lemma 2 and the triangle inequality. $\qquad\square$

We next show that vertex $\epsilon$-quasi isometry implies $\epsilon$-quasi isometry of simplexes.

*Proof of Lemma 3.* The following notations are used exclusively for this proof for convenience. We prove the lemma by induction on the number of vectors $k$, for two sets of vectors $\{z_1, \ldots, z_k\}$ and $\{z'_1, \ldots, z'_k\}$ that are the vertex $\epsilon$-qusi isometric. Let $\Delta = \text{conv}(\{z_1, \ldots, z_k\})$ and $\Delta' = \text{conv}(\{z'_1, \ldots, z'_k\})$.

The case $k = 2$ is trivial. For $k = 3$, consider $w_1, w_2 \in \Delta$. It suffices to show the case when $w_1, w_2$ are on two different sides of the triangle $\Delta$. For otherwise, we can always find a (smaller) triangle in $\Delta$ such that $w_1, w_2$ are on the two sides. Then without loss of generality, we assume that $w_1 = az_1 + (1-a)z_2$ and $w_2 = bz_1 + (1-b)z_3$ for some $a \geq b$. Let the corresponding vectors in $\Delta'$ be $w'_1$ and $w'_2$. Hence,

$$
\begin{aligned}
&\|(w_1 - w_2) - (w'_1 - w'_2)\| \\
=&\|(a-b)\big((z_1 - z_3) - (z'_1 - z'_3)\big) + (1-a)\big((z_2 - z_3) - (z'_2 - z'_3)\big)\| \\
\leq&(a-b)\|(z_1 - z_3) - (z'_1 - z'_3)\| + (1-a)\|(z_2 - z_3) - (z'_2 - z'_3)\| \\
\leq&(a-b)\epsilon + (1-a)\epsilon = (1-b)\epsilon \leq \epsilon.
\end{aligned}
$$

Notice that this implies the conclusion of the $\epsilon$-quasi isometry (by the triangle inequality), hence we have settled the case $k = 3$. Assume the claim holds for $2, \ldots, k$ and consider $k + 1$. For

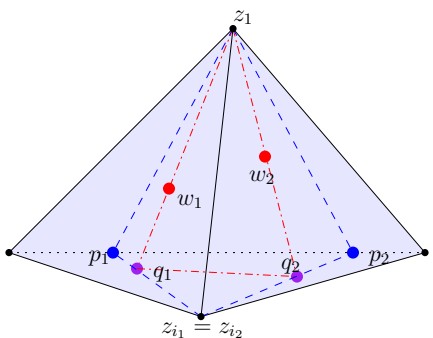

Figure 6: An illustration of the points $w_1, w_2, q_1, q_2, p_1, p_2, z_{i_1}, z_{i_2}$.

$w_1, w_2 \in \Delta$, we can always find a triangle with vertices $\{z_1, q_1, q_2\}$ with $q_1, q_2$ in the face $\Lambda_1 = \text{conv}(\{z_2, \ldots, z_{k+1}\})$ of $\Delta$ such that the following holds (illustrated in Fig. 6):

- Both $q_1$ and $q_2$ are on $\Lambda_1$.

- $w_1$ is on $[q_1, z_1]$ and $w_2$ is on $[q_2, z_1]$. Hence, $w_1$ (resp. $w_2$) is a convex combination of $q_1, z_1$ (resp. $q_2, z_1$).

- $q_1$ is on a line segment $[z_{i_1}, p_1]$, where $2 \le i_1 \le k+1$ and $p_1$ is in proper faces of $\Lambda_1$. Therefore, $p_1$ is a convex combination of at most $k$ vertices of $\Delta$ as $z_{i_1}$ is missing. The same holds for $q_2$ on a line segment $[z_{i_2}, p_2]$, $2 \le i_2 \le k+1$.

Correspondingly, we have $w_1', w_2', q_1', q_2', p_1', p_2', z_{i_1}', z_{i_2}'$ in $\Delta'$, with the exact same convex combination coefficients. To conclude $\|(w_1 - w_2) - (w_1' - w_2')\| \le \epsilon$, it suffices to successively apply the induction hypothesis to

- The proper face of $\Delta$ (resp. $\Delta'$) missing $z_{i_1}$ (resp. missing $z_{i_1}'$) and the proper face of $\Delta$ (resp. $\Delta'$) missing $z_{i_2}$ (resp. missing $z_{i_2}'$) to obtain

$$\|(z_1 - p_1) - (z_1' - p_1')\| \le \epsilon, \|(z_1 - p_2) - (z_1' - p_2')\| \le \epsilon.$$

- The triangle $\text{conv}(\{z_1, p_1, z_{i_1}\})$ (resp. $\text{conv}(\{z_1', p_1', z_{i_1}'\})$) and the triangle $\text{conv}(\{z_1, p_2, z_{i_2}\})$ (resp. $\text{conv}(\{z_1', p_2', z_{i_2}'\})$) to obtain

$$\|(z_1 - q_1) - (z_1' - q_1')\| \le \epsilon, \|(z_1 - q_2) - (z_1' - q_2')\| \le \epsilon.$$

- The proper face of $\Delta$ (resp. $\Delta'$) missing $z_1$ (resp. $z_1'$) to obtain

$$\|(q_1 - q_2) - (q_1' - q_2')\| \le \epsilon.$$

- The triangle $\text{conv}(\{z_1, q_1, q_2\})$ (resp. $\text{conv}(\{z_1', q_1', q_2'\})$) to obtain

$$\|(w_1 - w_2) - (w_1' - w_2')\| \le \epsilon.$$

The lemma is proved. $\qquad\square$

We are now ready to prove the main results of the paper.

*Proof of Theorem 1.* For (a), recall $N_{v,c}$ is the set of neighbors of $v$ whose label is $c$. We estimate $\|y_v - e_{N_v}\|$ as follows (notice terms with $|N_{v,c}| = 0$ are ignored):

$$\|y_v - e_{N_v}\|^2 = \|\sum_{c \in C} \frac{|N_{v,c}|}{|N_v|} e_c - \sum_{c \in C} \frac{1}{|N_v|} \sum_{v_i \in N_{v,c}} x^i\|^2$$

$$\leq (\sum_{c \in C} \frac{|N_{v,c}|}{|N_v|} \|e_c - \frac{1}{|N_{v,c}|} \sum_{v_i \in N_{v,c}} x^i\|)^2$$

$$\leq (\sum_{c \in C} \frac{|N_{v,c}|}{|N_v|})(\sum_{c \in C} \frac{|N_{v,c}|}{|N_v|} \|e_c - \frac{1}{|N_{v,c}|} \sum_{v_i \in N_{v,c}} x^i\|^2) \text{ by Cauchy-Schwartz}$$

$$= \sum_{c \in C} \frac{|N_{v,c}|}{|N_v|} \|e_c - \frac{1}{|N_{v,c}|} \sum_{v_i \in N_{v,c}} x^i\|^2.$$

We apply Lemma 2, there is a $K$ independent of $v$ such that

$$\mathbb{P}\left(\|e_c - \frac{1}{|N_{v,c}|} \sum_{v_i \in N_{v,c}} x^i\| \leq (\frac{\delta|N_v|}{|N_{v,c}|})^{1/2}\right)$$

$$\geq 1 - \exp(-K|N_{v,c}| \cdot \frac{\delta|N_v|}{|N_{v,c}|} + 1/4)$$

$$= 1 - \exp(-K\delta|N_v| + 1/4).$$

Therefore, with probability at least $(1 - \exp(-K\delta|N_v| + 1/4))^{|C|}$, we have

$$\|y_v - e_{N_v}\|^2 \leq \sum_{c \in C} \frac{|N_{v,c}|}{|N_v|} \cdot \frac{\delta|N_v|}{|N_{v,c}|} = |C|\delta.$$

This proves the claim.

To prove (b), it suffices to prove $\|e_c - e_{N_v}\| \geq rd_{v,c} - \epsilon$, and then the claim follows from (a) by using the triangle inequality. By Lemma 3, the condition on the vertex $\epsilon$-quasi isometry implies that $\Delta_e \approx_\epsilon \Delta$. Let $u \in \Delta$ be the vector that corresponds to $e_{N_v} \in \Delta_e$, i.e., $u = \sum_{c \in C} \frac{|N_{v,c}|}{|N_v|} u_c$. As $\Delta$ is a regular simplex of size length $r$, $\|u - u_c\| = rd_{v,c}/\sqrt{2}$. This can be derived from the probability simplex, which is a regular simplex of side length $\sqrt{2}$. By $\epsilon$-quasi isometry, we have $\|e_c - e_{N_v}\| \geq \|u - u_c\| - \epsilon = rd_{v,c}/\sqrt{2} - \epsilon$. The result follows. $\qquad \square$

*Proof of Theorem 2.* Let $\mathcal{V}_1$ (resp. $\mathcal{V}_2$) be the subspace of the feature space spanned by $\{e_c \mid c \in C\}$ (resp. $\{z_c \mid c \in C\}$). Recall the vertex set of the probability simplex $\Delta_C$ in $\mathbb{R}^{|C|}$ is $\{p_c \mid c \in C\}$. We define a linear transformation $L_1 : \mathbb{R}^{|C|} \to \mathcal{V}_1$ induced by $p_c \mapsto e_c, c \in C$. Similarly, define $L_2 : \mathbb{R}^{|C|} \to \mathcal{V}_2$ induced by $p_c \mapsto z_c, c \in C$. Let $\sigma_1$ (resp. $\sigma_2$) be the least singular value of $L_1$ (resp. $\sigma_2$). Order $C$ by $c_1, \ldots, c_l, l = |C|$, then the columns of $L_1$ (resp. $L_2$) are $\{e_{c_1}, \ldots, e_{c_l}\}$ (resp. $\{z_{c_1}, \ldots, z_{c_l}\}$).

As $\Delta = \mathrm{conv}(\{0\} \cup \{z_c \mid c \in C\})$ is an $(r_1, r_2)$-regular simplex, up to rotation, $\Delta$ can be positioned as follows:

- In $\mathcal{V}_2$, first form $\Delta' = \{z'_c \mid c \in C\}$, where each $z'_c$ is on a unique (positive) coordinate axis with the only non-zero coordinate $r_1/\sqrt{2}$.

- $z'_0$ is the unique point placed below $\Delta'$ such that $\mathrm{conv}(\{z'_0, z'_c \mid c \in C\})$ is $(r_1, r_2)$-regular.

- $\Delta = \mathrm{conv}(\{z'_0, z'_c \mid c \in C\}) - z'_0$.

Notice that rotation does not change the size of the determinant and singular values of $L_2$. For the above special $\Delta$, the determinant of $L_2$ has size $vol(\Delta)$ and a singular value of $L_2$ is either $r_2\sqrt{k}$

(whose eigenvector is $(1, \ldots, 1)^{\mathsf{T}}$) or $\left(vol(\Delta)/(r_2\sqrt{k})\right)^{1/(k-1)}$. Recall that the volume of $\Delta$ is

$$vol(\Delta) = \frac{\sqrt{k}r_1^{k-1}r_2}{k!\sqrt{2^{k-1}}}.$$

Therefore, the constant $\rho(r_1, r_2)$ is the least singular value $\sigma_2$ of $L_2$.

Let $p = \sum_{1 \leq i \leq l} a_i p_{c_i} \neq 0$ be a vector in $\mathcal{V}_1$ that realizes $\sigma_1$ of $L_1$, i.e., $\sum_{1 \leq i \leq l} a_i^2 = 1$ and $\|L_1(p)\| = \sigma_1$. By the Cauchy-Schwartz inequality, $t = \sum_{1 \leq i \leq l} |a_i|$ satisfies $t^2 \leq l = |C|$ and hence $t \leq \sqrt{|C|}$.

Assume without loss of generality that $a_1, \ldots, a_k \geq 0$ and $a_{k+1}, \ldots, a_l < 0$. We estimate

$$\frac{1}{t}\sigma_1 = \frac{1}{t}\|\sum_{1 \leq i \leq l} a_i L_1(p_{c_i})\| = \frac{1}{t}\|\sum_{1 \leq i \leq l} a_i e_{c_i}\|$$

$$= \|\sum_{1 \leq i \leq k} \frac{a_i}{t} e_{c_i} - \sum_{k+1 \leq j \leq l} \frac{a_j}{t} e_{c_j}\|.$$

As $\sum_{1 \leq i \leq k} \frac{a_i}{t} \leq 1$ and $\sum_{k+1 \leq j \leq l} \frac{a_j}{t} \leq 1$, both vectors $\sum_{1 \leq i \leq k} \frac{a_i}{t} e_{c_i}$ and $\sum_{k+1 \leq j \leq l} \frac{a_j}{t} e_{c_j}$ are in $\mathrm{conv}(\{0\} \cup \{e_c \,|\, c \in C\})$. By vertex $\epsilon$-quasi isometry of $\{0\} \cup \{e_c \,|\, c \in C\}$ with $\{0\} \cup \{z_c \,|\, c \in C\}$, we have

$$\|\sum_{1 \leq i \leq k} \frac{a_i}{t} e_{c_i} - \sum_{k+1 \leq j \leq l} \frac{a_j}{t} e_{c_j}\|$$

$$\geq \|\sum_{1 \leq i \leq k} \frac{a_i}{t} z_{c_i} - \sum_{k+1 \leq j \leq l} \frac{a_j}{t} z_{c_j}\| - \epsilon$$

$$= \frac{1}{t}\|L_2(p)\| - \epsilon \geq \frac{1}{t}\sigma_2 - \epsilon.$$

Therefore, we have

$$\sigma_1 \geq \sigma_2 - t\epsilon \geq \sigma_2 - \sqrt{|C|}\epsilon = \rho(r_1, r_2, \epsilon)^{-1} > 0.$$

This implies that $L_1$ is invertible and the operator norm $\|L_1^{-1}\|$ is bounded by $\rho(r_1, r_2, \epsilon)$. We finally define $L$ to be $L_1^{-1}$ on $\mathcal{V}_1$ and $0$ on the orthogonal complement of $\mathcal{V}_1$.

It is now easy to prove the two claims. For (a), it is easy to verify that $\widehat{y}_c = (\sum_{v_i: \text{ class } c} L(x^i))/n$. Hence,

$$\|p_c - \widehat{y}_c\| = \|L(e_c) - L(g_c)\| \leq \rho(r_1, r_2, \epsilon)\|e_c - g_c\|.$$

Therefore, (a) follows from Lemma 2. Similarly, for (b), we have

$$\|L(y_v) - \mu_v\| = \|L(y_v) - L(e_{N_v})\| \leq \rho(r_1, r_2, \epsilon)\|y_v - e_{N_v}\|.$$

The claim in (b) follows from Theorem 1. $\qquad\square$

*Proof of Corollary 1.* Let the vectors in $\Delta$ that corresponds to $e_{N_v}$ and $\overline{e_{N_v}}$ be $u$ and $\overline{u}$ respectively. As the simplex $\Delta$ is degenerate, we have $u = \overline{u}$. Therefore, by the vertex $\epsilon$-isometry and Lemma 3, we have $\|e_{N_v} - \overline{e_{N_v}}\| \leq \epsilon$. The corollary follows immediately from Theorem 1 (a). $\qquad\square$

*Proof of Lemma 4.* To show $M$ is stochastic, we compute for each $c \in C$:

$$\sum_{c' \in C} \alpha_{c,c'} = \sum_{c' \in C} \frac{1}{n_c} \sum_{v_i \in D_c} \frac{|N_{v_i, c'}|}{|N_{v_i}|} = \frac{1}{n_c} \sum_{v_i \in D_c} \sum_{c' \in C} \frac{|N_{v_i, c'}|}{|N_{v_i}|} = \frac{1}{n_c} \sum_{v_i \in D_c} 1 = 1.$$

As each node has a self-connection, $M$ is aperiodic. For any proper subset $C'$ of $C$, the set of nodes $\cup_{c \in C'} D_c$ has some neighbors whose class labels do not belong to $C'$. For otherwise, the proper subset $\cup_{c \in C'} D_c$ is a union of connected components of $G$, which contradicts the assumption that $G$ is connected. This implies that any $(C \backslash C') \times C'$ subblock of $M$ contains some positive entries. Therefore, $M$ is irreducible. Being aperiodic and irreducible, $M$ has a unique stationary distribution (see Lalley (2016)). $\qquad\square$

## C    RELATED WORKS

**The proposed tricks**    The tricks proposed in Section 5 belong to the following generic categories of machine learning techniques: *rewiring*, *early stopping*, and *feature normalization*. However, different research works propose different realizations of the above categories of machine learning techniques. The theoretical motivation and the explicit procedure differentiate approaches belonging to the same category.

For "rewiring", DropEdge (Rong et al., 2020) proposes to randomly drop a fraction of edges in each iteration to alleviate the side-effects of oversmoothing. Bober et al. (2023) proposes to add or remove edges to change the graph curvature for issues such as over-squashing. Ji et al. (2023a) considers random edge removal at estimated class boundaries to (relatively) enhance intra-class connectivities. In contrast to these earlier works, we propose rewiring by adding edges to reduce feature variance. The trick is derived directly from our theoretical findings as discussed in Section 5 (see the subsection on Simple tricks (1)).

"Early stopping" is usually used to reduce overfitting (Prechelt, 2012) or as a regularization in boosting (Jiang, 2004). In our paper, we propose *extremely early* stopping (for heterophilic graphs). It is based on the idea that we want to prevent the aggregated features of training nodes from fitting to their respective label classes, which is unlike what people usually do. This is because the aggregated features are likely to be close to the mixed centroids as we have discussed in Section 4.

For "feature normalization", people usually do so in general machine learning for model stability and efficiency, and to standardize different scales when features are measured (Singh & Singh, 2020; Subasi, 2020). For GNN, for example, Zhao & Akoglu (2020) (the PairNorm model) proposes a post-processing normalization layer in terms of the total pairwise squared distance introduced in the paper, to alleviate oversmoothing. However, *initial feature realization* is rarely implemented in GNNs. We propose doing so for the very specific reason of reshaping the simplicial complex $\Delta_g$ (supported by Theorem 2 and Corollary 1).

**Geometry and GNNs**    Recent works have used more complicated topological and geometric tools to model node correlations or to capture hidden structural network information. For example, Horn et al. (2022) applies topological data analysis to retrieve global topological information such as the number of cycles to enhance GNN expressiveness. The survey Papillon et al. (2024) contains a comprehensive overview of models when graphs are replaced with more complicated and expressive structures such as cellular complexes and hypergraphs. On the other hand, Han et al. (2024) on geometric graph neural networks summarizes approaches given additional geometric information such as explicit locations of nodes in a 3D space. They put emphasis on geometric invariance and they are particularly useful for bio-chemical datasets. Unlike these works, our theme is on the geometry of the features instead of the network structure. Hence, though geometric and topological tools are employed, the subjects of study are different.

## D    DATASETS AND IMPLEMENTATION DETAILS

### D.1    DATASETS

The following datasets are used and studied at various places in the paper (including the appendices): Cora, Citeseer, PubMed, Ogbn-arxiv, Texas, Cornell, Wisconsin, Chameleon, Squirrel, Actor, Penn94, arXiv-year, and genius (see Lim et al. (2021)). Their statistics are shown in Table 6.

### D.2    HYPERPARAMETERS

There are two hyperparameters used in the tricks in Section 5: $\eta$ for the edge addition probability and $\mathcal{E}$ for the number of epochs in training. They are tuned according to the following general procedure. For $\eta$, we will first consider $\eta = 1, 0.5, 0.2, 0.1, 0.001, 0.0001$ and then fine-tune around one of these values using the validation set. We also keep track of the number of edges as a useful reference when tuning, as graphs for different datasets may have large difference in edge size. For example, the Texas graph has only 295 edges and an average degree of $\approx 3.2$ while the Squirrel graph has $217k$ edges and an average degree of $\approx 83.5$.

Table 6: Dataset statistics

| Dataset | Nodes | Edges | Classes | Node Features | Data splits |
|---|---|---|---|---|---|
| Cora | 2708 | 5429 | 7 | 1433 | standard |
| Citeseer | 3327 | 4732 | 6 | 3703 | standard |
| PubMed | 19717 | 88651 | 3 | 500 | standard |
| Ogbn-arxiv | 169343 | 1166243 | 40 | 128 | standard |
| Texas | 183 | 295 | 5 | 1703 | 48%/32%/20% |
| Cornel | 183 | 280 | 5 | 1703 | 48%/32%/20% |
| Wisconsin | 251 | 466 | 5 | 1703 | 48%/32%/20% |
| Chameleon | 2277 | 36101 | 5 | 2305 | 48%/32%/20% |
| Squirrel | 5201 | 217073 | 5 | 2089 | 48%/32%/20% |
| Actor | 7600 | 33391 | 5 | 932 | 48%/32%/20% |
| Penn94 | 41554 | 1362229 | 2 | 4814 | 50%/25%/25% |
| arXiv-year | 169343 | 1166243 | 5 | 128 | 50%/25%/25% |
| genius | 421961 | 984979 | 2 | 12 | 50%/25%/25% |

For the number of epochs, let $\mathcal{E}_0$ be the number of epochs of the base model ($\mathcal{E}_0 = 1000$ for CDE and GloGNN and $= 200$ for other base models). We consider $\mathcal{E} = \mathcal{E}_0/20, \mathcal{E}_0/10, \mathcal{E}_0/4, \mathcal{E}_0/2$ and then fine-tune around one of these values.

### D.3 IMPLEMENTATION

The tricks are simple and thus straightforward to implement. The source code to reproduce our results can be found at `https://github.com/YananZhao0630/M-AE-M-AEN` (base model: ACM-GCN). Experiments are performed on a workstation with a single NVIDIA GeForce RTX 3090 GPU and 24GB memory.

## E NUMERICAL EVIDENCE AND DISCUSSIONS

In this appendix, we provide numerical evidence with discussions to support various claims of the paper.

### E.1 CHOICE OF THE MODEL FOR ANALYAIS

In Table 7, we show results from different variants of GCN on Cora and Citeseer datasets, to justify our choice of single-layer convolution without ReLU for theoretical analysis in Section 3.

Table 7: Test accuracies of GCN variants on Cora and Citeseer (C: Convolution layer, R: ReLU)

| | CRCR | CRC | CC | CR | C |
|---|---|---|---|---|---|
| Cora | 71.7% | 80.3% | 81.1% | 77.9% | 77.6% |
| Citeseer | 63.7% | 70.8% | 71.8% | 66.7% | 68.4% |

### E.2 THE SHAPES OF $\Delta_g$

In Fig. 7, we show the pairwise Euclidean distance of $\{g_c \,|\, c \in C\}$ for all the datasets in Appendix D.1. In addition, we also show the distance from each $g_c$ to the origin (cf. Theorem 2).

From Fig. 7, we may have a rough estimation of how each $\Delta_g$ deviates from a regular simplex. This is measured by a pair $(\epsilon, r)$ such that $\Delta_g$ is vertex $\epsilon$-quasi isometric to a regular simplex of side length $r$. For simplicity, we compute the proxies $(\max_{\Delta_g} - \min_{\Delta_g})/2$ and $(\max_{\Delta_g} + \min_{\Delta_g})/2$ for $\epsilon$ and $r$ respectively, where $\max_{\Delta_g}$ (resp. $\min_{\Delta_g}$) is the maximum (resp. minimum) among all pairwise

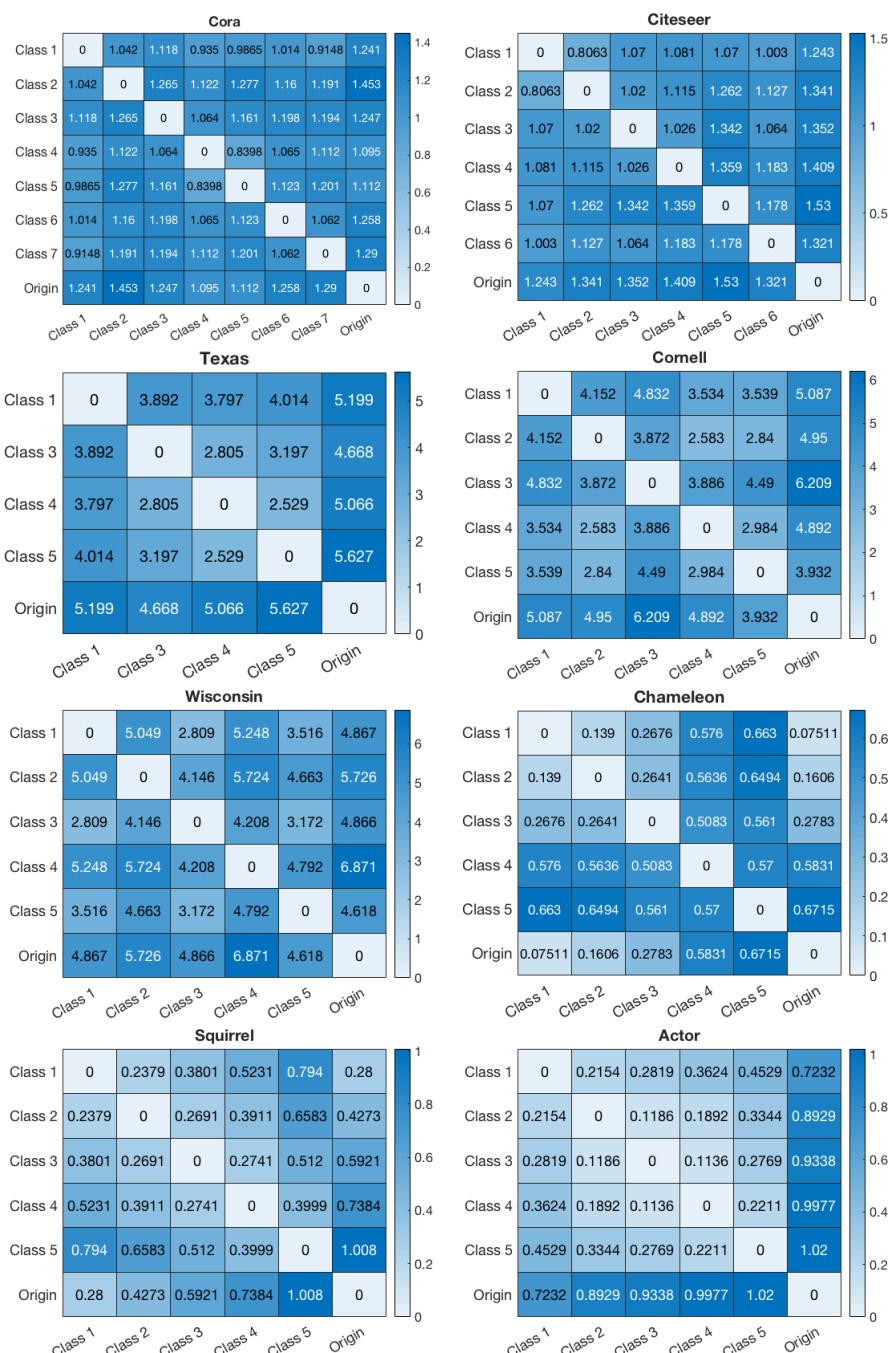

Figure 7: The pairwise distance of $\{g_c \mid c \in C\}$ for all the datasets. For the Texas dataset, class 2 is omitted as there is only 1 node from class 2. The last row/column contains the distance of each $g_c$ to 0.

vertex distances of $\Delta_g$. In Table 8, we tabulate their ratio $\tau$ as a proxy for $\epsilon/r$. The smaller its value is, the more regular the shape of $\Delta_g$ is.

We see that for Chameleon, Squirrel, and Actor, the estimated $\tau$ is $\approx 0.5$ or larger, which indicates their $\Delta_g$'s are much less regular. Moreover, from Fig. 7, we can sketch the shape of each $\Delta_g$. From the illustration in Fig. 8, each $\Delta_g$ has sides with relatively small lengths or side triangles with small

Table 8: The estimated value $\tau$ for each dataset.

| Cora | Citeseer | Texas | Cornel | Wisconsin | Chameleon | Squirrel | Actor |
|------|----------|-------|--------|-----------|-----------|----------|-------|
| 0.174 | 0.255 | 0.177 | 0.236 | 0.342 | 0.653 | 0.494 | 0.599 |

areas. Therefore, they resemble degenerate simplexes discussed in Section 3.3. This explains why GNN models generally perform poorly on these datasets by Corollary 1.

A more general notion of degeneracy (as compared with that defined in Section 3.3) is that some $u_c$ belongs to the convex hull of $\{u_{c'} \mid c' \neq c\}$ (this notion describes better the shape of Actor). If $\Delta_e$ is $\epsilon$-quasi isometric to a degenerate simplex in this sense, then similar to Corollary 1, it will be hard to identify a feature, that corresponds to the label class $c$, unambiguously.

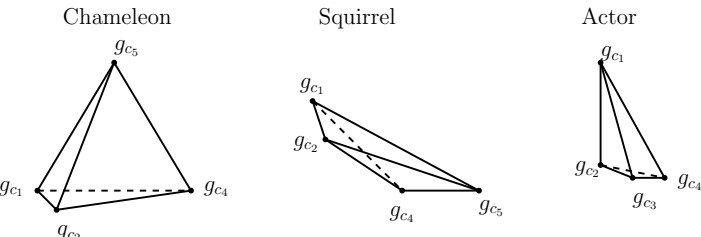

Figure 8: We show the illustrations of a part of $\Delta_g$ for Chameleon, Squirrel and Actor datasets. For visualization, we can only show the tetrahedron formed from 4 vertices.

We offer a plausible explanation of the above phenomenon on the shapes of $\Delta_g$. For each dataset, we compute and tally in Fig. 9 the 1-norm (i.e., number of non-zero components) of the original feature of each node. From Fig. 9, we see that for Chameleon, Squirrel, and Wisconsin, node feature sizes are much smaller than those of other datasets on average. Consequently, centroids can be very close to 0 and hence close to each other, as observed in Fig. 7. Intuitively, this suggests that for these datasets, the features are not sufficiently expressive.

### E.3 THE USEFULNESS OF KNOWING $\mu_v$

To numerically verify the discussions following Theorem 2, we consider a thought experiment by assuming that $\mu_v$ of each $v$ is known (caution: this is hypothetical and the information is unavailable in the actual problem). Let $\nu_v$ be the output label distribution of a GCN model. We add the regularization term $\frac{\eta}{\sqrt{n}} \sum_{v \in V} \|\nu_v - \mu_v\|$ to the loss $\ell$ of the model, where $\eta$ is the regularization weight. We vary $\eta$ and the test accuracies are shown in Fig. 10. From the results, we see that the regularization term indeed plays a dominant role. Moreover, a 1-layer GCN is better than a 2-layer GCN. This is due to that a 1-layer GCN considers exactly the 1-hop neighbor of each node. The observations indeed verify the conclusion of Theorem 2 (b). Of course in theory, if the normalized adjacency matrix is invertible, then knowing the ground truth label of each node is equivalent to knowing $\mu_v$ for each node.

### E.4 AGGREGATED FEATURES AND CENTROIDS

To understand the relation between node features and centroids, we perform the following study. Fix a label class $c$. We randomly choose 5 nodes from each class and find the distance from their average features to $g_c$. This process is repeated for 60 instances. In addition, for the class $c$, we also obtain results for the average over 10 random nodes and over 20 random nodes. From Fig. 11, we see that for Cora, Citeseer, Texas, Cornell, and Wisconsin datasets, averaging over nodes with label $c$ is closer to $g_c$. However, the same is not observed in Chameleon, Squirrel, and Actor datasets. For example, averaging features of 10 random nodes with label $c$ is not necessarily closer to $g_c$ than averaging features of 5 random nodes from a different class. A possible reason is that different centroids for these datasets can be very close to each other (e.g., $(g_{c_1}, g_{c_2})$ for Chameleon, $(g_{c_2}, g_{c_3})$ for Squirrel

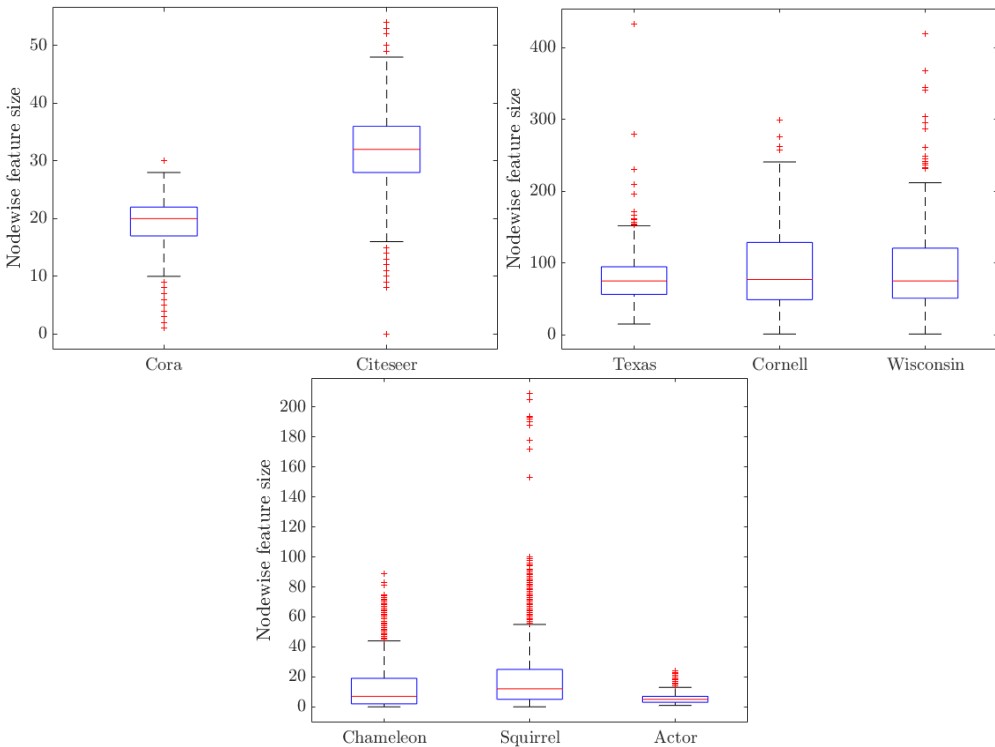

Figure 9: The norm (i.e., number of non-zero components) of the original feature of each node.

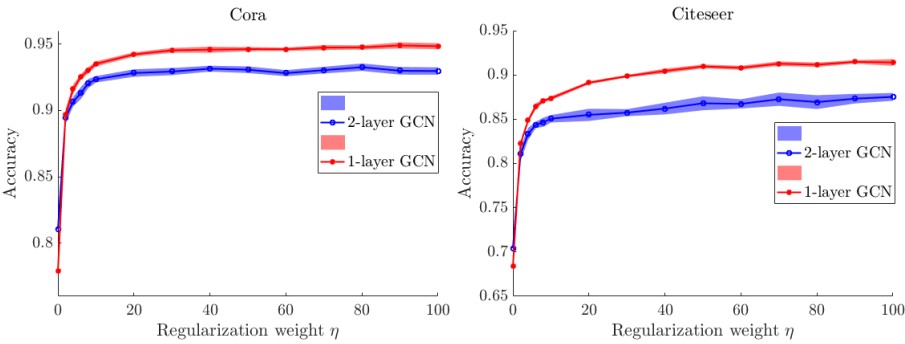

Figure 10: A regularization scheme that uses neighborhood label distributions.

and $(g_{c_2}, g_{c_3})$ for Chameleon for actor). The observation is consistent with our discussions regarding the shapes of $\Delta_g$.

We investigate the features aggregated from a single-layer graph convolution. For each dataset, we randomly choose 50 nodes. For each choose $v_i$ with label $v_i$, we compute its aggregated feature $y_{v_i}$, and find its Euclidean distance to the ground-truth $g_{c_i}$, the mixed centroid $e_{N_{v_i}}$ and the centroid of a random wrong class. From the results shown in Fig. 12, we see that for the homophilic datasets Cora and Citeseer, the distance to the mixed centroid is almost the same as that to the true label centroid, and both are smaller than the distance to a wrong label centroid. The former observation reflects the homophilic property. For Texas, Cornell, and Wisconsin, as predicted by Theorem 2, the distance to the mixed centroid is noticeably smaller than that to the true label centroid. On the other hand, consistent with our earlier discussions on Chameleon, Squirrel, and Actor, the three types of differences are all very similar, rendering classification difficult measured either in both accuracy or $\ell_v$ (see Section 5).

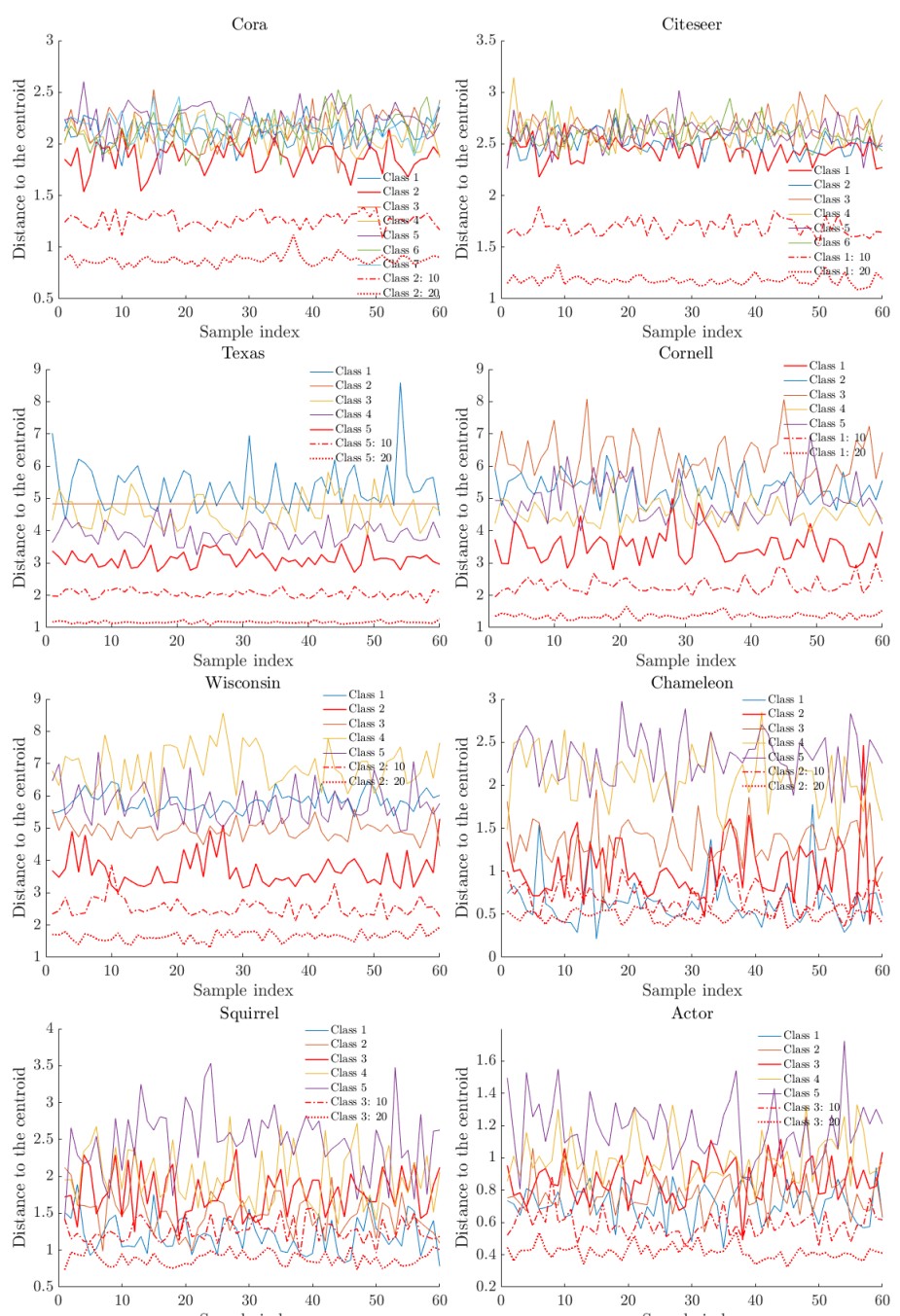

Figure 11: Euclidean distances from average features to class centroids. We only show the results for one fixed $c$. Generally, a similar pattern is observed for other classes given the same dataset. In the legends, ":10" or ":20" means average over 10 or 20 sample respectively.

### E.5 ERRORS AND CENTROID DISTANCES

Our theory suggests a strong (negative) correlation between test error and the distance between class centroids. More specifically, for each class $c_i$, we count the number of test nodes with ground-truth label $c_i$ while wrongly predicted as $c_j$, for each $j \neq i$. The results are shown in Fig. 13. If we look at each row, say the $i$-th row of the tables, then we observe the following general pattern. Entries with relatively larger error count for that row usually correspond to smaller centroid distances for the same

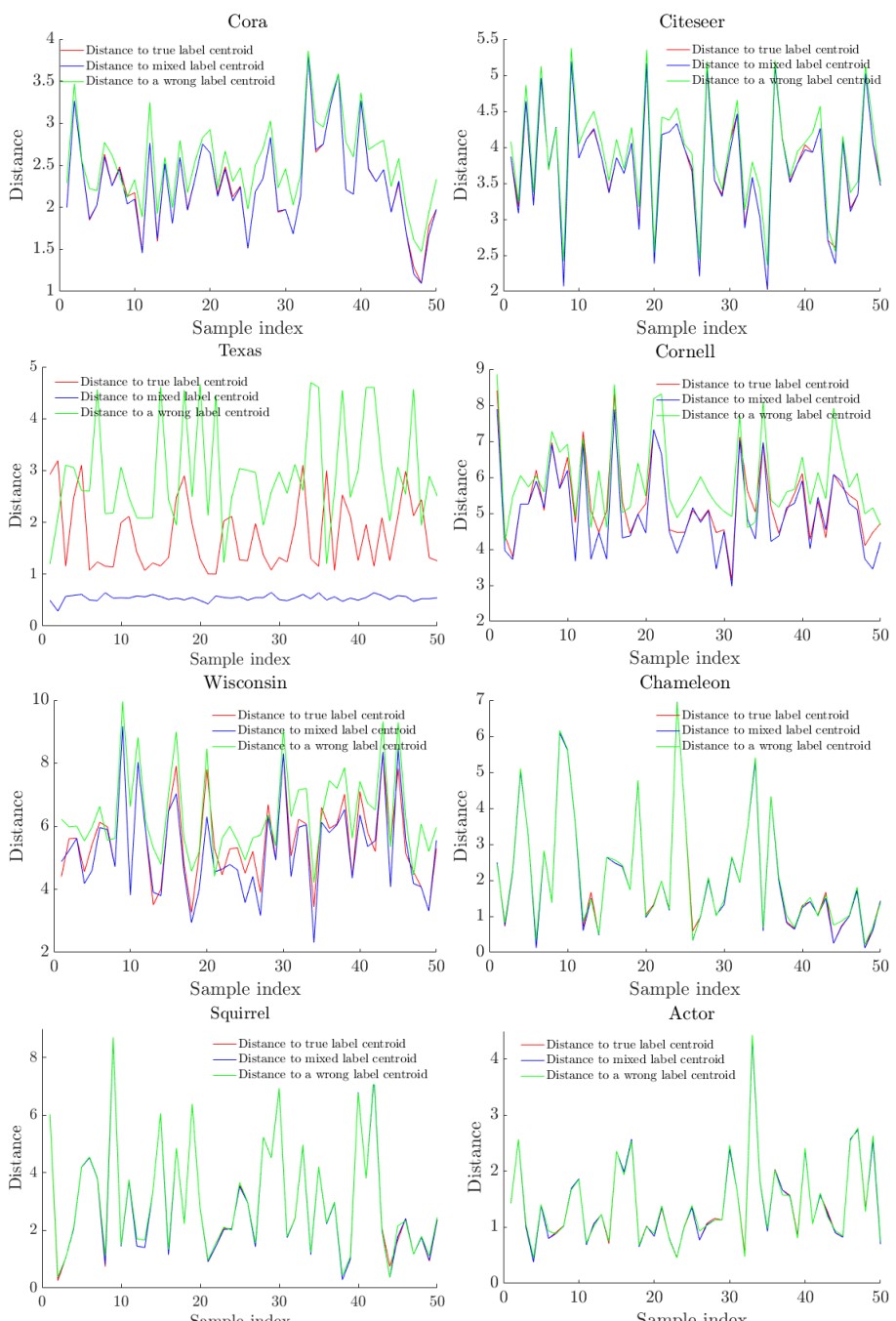

Figure 12: We show the distance from $y_{v_i}$ to its ground-truth label centroid, the mixed label centroid, and a random wrong label centroid, for 50 randomly chosen $v_i$.

row (right column of Fig. 13). (Though it is not true for every single instance.) This is consistent with our theory.

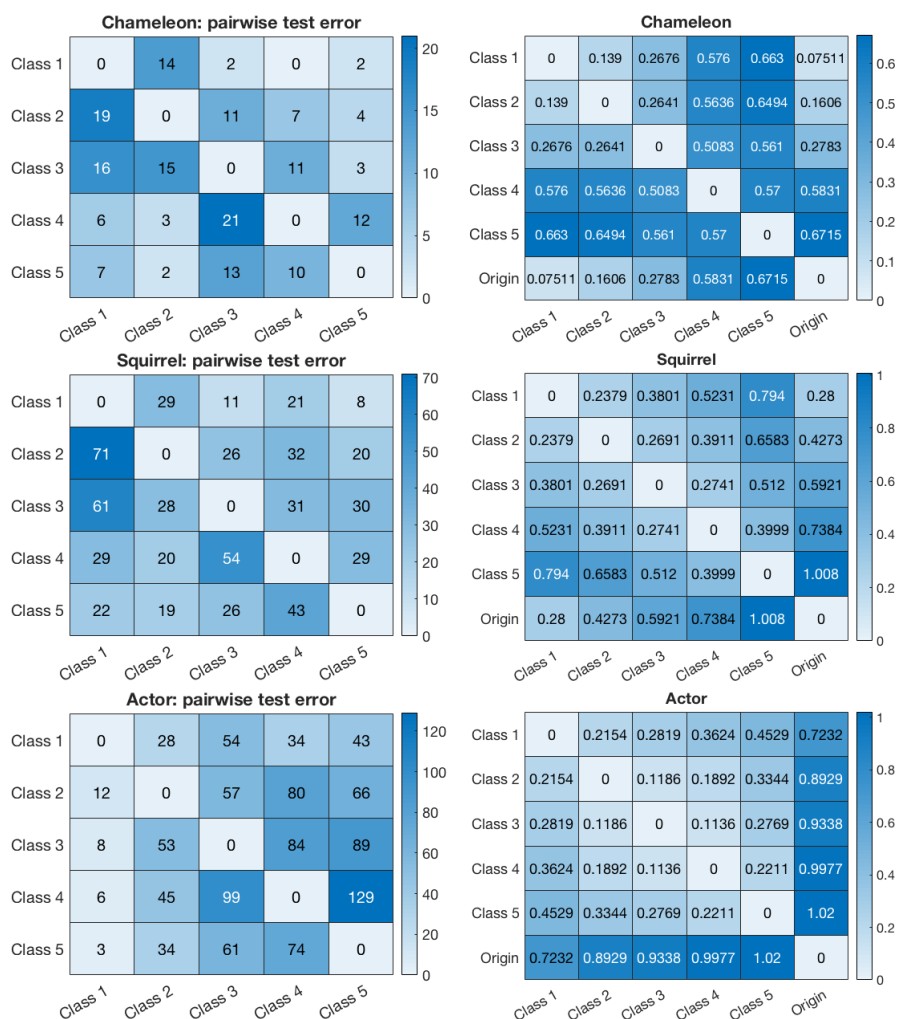

Figure 13: In the left column, we show the summary of test error for each dataset. In each table, the $(i, j)$-th entry is the number belonging to the error type that predicts label class $c_i$ as class $c_j$. The right column is taken from Fig. 7 for comparison.

### E.6 FEATURE NORMALIZATION

We analyze the normalization trick (Section 5) when applied to Chameleon, Squirrel, and Actor datasets. As the features are changed, we re-compute the resulting pairwise distance of the centroids $\{g_c \mid c \in C\}$ (cf. Fig. 7).

Comparing Fig. 7 and Fig. 14, it is observed that the resulting $\Delta_g$ becomes more regular. To quantify, for the three datasets, we compute the value $\tau$ (cf. Table 8) and find that it reduces from approximately $0.653, 0.494$ and $0.599$ to $0.497, 0.284$ and $0.592$ respectively. This is consistent with the performance improvement seen in Section 5 Table 3. Moreover, we notice that for the Actor dataset, the change in $\tau$ is small, and so is the corresponding performance improvement for the vanilla 1-layer GCN model.

In Fig. 14, we also show the distance from average features to the centroid of each class (cf. Fig. 11). Comparing Fig. 11 with Fig. 14, we see that after feature normalization, the average features of a class $c_i$ are closer (to its centroid $g_{c_i}$) than the average features of a different class. The observation is more prominent for the Chameleon and Squirrel datasets, accounting for the effectiveness of normalization for these datasets.

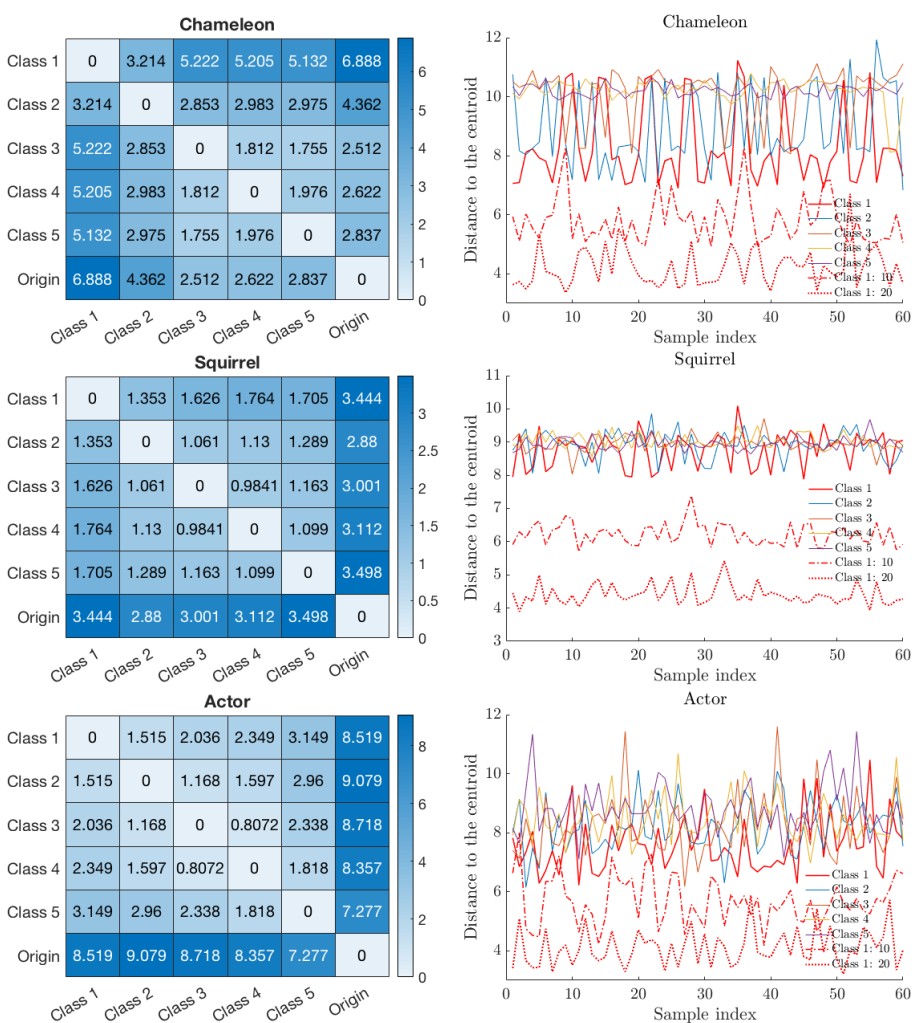

Figure 14: We show the pairwise pairwise distance of $g_c, c \in C$ for the datasets after normalization (cf. Fig. 7). Moreover, we also show figures analogous to Fig. 11 after normalization.

# F MORE RESULTS

## F.1 RESULTS ON MORE DATASETS

**Homophilic graphs** In Table 9, we show the classification results for homophilic graphs. The proposed tricks generally improve the performance of base models. However, the tricks have a stronger impact on the heterophilic datasets.

For the Ogbn-arxiv dataset, we see that the best accuracy among all models is $\approx 60\%$. According to our discussion, the dataset is likely to be intrinsically difficult by having an almost degenerate $\Delta_g$. We verify that this is indeed the case, as its estimated $\tau$ value (cf. Table 8) is $\approx 0.67$. Based on our discussion in Section 5, the normalization trick might be useful to enhance the performance, and it is indeed true as shown in Table 10.

**Large scale non-homophilous graphs** In Table 11, we show the node classification results for a few large scale non-homophilous graphs introduced in Lim et al. (2021). We use the model LINKX proposed in Lim et al. (2021) as the based model and apply jointly the edge addition and early stopping tricks described in Section 5. We see an improvement in the test accuracy for each dataset.

Table 9: Node classification results(%) for homophilic graphs. "-I" and "-II" stand for 1 and 2 layers respectively. For the base models, we use directly the source code and data split by the respective authors. Suffix "-AE" is for the models with edge addition and early stopping tricks.

| Method | Cora | Citeseer | PubMed | Ogbn-arxiv |
|---|---|---|---|---|
| GCN-I | 75.64±0.53 | 70.66±0.84 | 76.97±0.18 | 42.54±0.88 |
| GCN-I-AE | 76.29±0.37 | 71.03±0.72 | 77.00±0.15 | 45.35±1.26 |
| GCN-II | 80.95±0.41 | 70.91±0.48 | 79.29±0.23 | 37.72±0.55 |
| GCN-II-AE | 81.19±0.69 | 71.32±0.24 | 79.96±0.25 | 44.72±0.40 |
| GAT-I | 63.50±0.83 | 54.98±4.30 | 66.54±3.17 | 52.02±1.46 |
| GAT-I-AE | 64.48±0.64 | 57.18±2.33 | 69.74±4.43 | 52.21±1.07 |
| GAT-II | 73.44±2.50 | 67.82±1.46 | 75.99±0.98 | 53.40±0.36 |
| GAT-II-AE | 73.73±2.21 | 68.36±0.52 | 76.12±0.68 | 53.63±0.51 |
| ACM-GCN | 84.19±0.02 | 75.42±0.02 | 89.07±0.02 | 58.84±1.00 |
| ACM-GCN-AE | 84.63±0.01 | 76.95±0.02 | 90.30±0.01 | 59.01±1.24 |
| GraphCON | 87.44±1.35 | 75.10±2.37 | 87.20±0.88 | 60.43 ± 1.21 |
| GraphCON-AE | 87.91±1.40 | 75.41±2.02 | 87.38 ±0.66 | 60.67 ± 0.72 |
| CDE | 83.80±1.19 | 73.45±1.60 | 89.80±0.28 | OOM |
| CDE-AE | 83.88±0.88 | 73.72±1.72 | 89.85±0.25 | OOM |
| GloGNN | 86.22±1.17 | 75.22±1.83 | 87.79±0.22 | 60.00±0.33 |
| GloGNN-AE | 86.52±1.23 | 75.26±1.61 | 87.97±0.26 | 60.24±0.22 |

Table 10: Normalization (suffix: "-AEN") is applied to ACM-GCN for Ogbn-arxiv.

| Method | ACM-GCN | ACM-GCN-AE | ACM-GCN-AEN | % ↑ over $\mathcal{M}$ |
|---|---|---|---|---|
| Ogbn-arxiv | 58.84±1.00 | 59.01±1.24 | 66.63±0.48 | 13.2% |

**Limited feature information** We study the performance of our proposed tricks for small feature sizes by randomly removing a fraction of features. We choose the best base model ACM-GCN and consider retaining $1/2, 1/4, 1/8$ feature dimensions. The results are shown in Table 12 and we see that our proposed tricks generally improve the performance.

Table 11: Node classification results(%) for large scale non-homophilous graphs. Suffix "-AE" is for the model with tricks.

| Method | arXiv-year | Penn94 | genius |
|---|---|---|---|
| LINKX | 54.05±0.36 | 83.79±0.59 | 90.54±0.22 |
| LINKX-AE | 62.60±0.21 | 84.56±3.69 | 94.61±0.16 |

Table 12: Node classification results(%) if a fraction of feature dimension is retained.

| Feature dimension | Full | 1/2 | 1/4 | 1/8 |
|---|---|---|---|---|
| Chameleon | | | | |
| ACM-GCN | 71.29±8.06 | 61.95±7.13 | 52.70±5.36 | 44.17±3.87 |
| ACM-GCN-AEN | 75.22±9.99 | 63.53±7.37 | 56.09±5.36 | 46.07±4.92 |
| Squirrel | | | | |
| ACM-GCN | 55.07±8.96 | 49.67±7.43 | 42.10±5.08 | 36.27±2.72 |
| ACM-GCN-AEN | 59.47±11.41 | 50.49±7.81 | 44.10±5.93 | 37.67±2.38 |
| Actor | | | | |
| ACM-GCN | 37.28±2.76 | 33.74±2.34 | 31.32±1.67 | 26.89±0.97 |
| ACM-GCN-AEN | 47.62±6.38 | 35.09±1.91 | 32.33±1.53 | 29.23±0.85 |

## F.2 Conformal prediction

As we have argued in Section 5, we may consider estimating the neighborhood label distribution $\mu_v$ of each node $v$ as an alternative objective. We have seen that the tricks proposed in Section 5 improve the estimated $\mu_v$ for Texas, Cornell, and Wisconsin datasets. We have also provided possible reasons regarding why improvements are not seen for Chameleon, Squirrel, and Actor datasets.

On the other hand, *conformal prediction* for GNNs has recently drawn much attention (Clarkson, 2023; Zargarbashi et al., 2023; Huang et al., 2023). In a nutshell, it removes each class with a very small predicted probability, based on a calibration node set. We view each removed class as "noise", and thus conformal prediction is viewed as a denoising process. More specifically, for each test node $v$, let $\nu_v$ be its predicted probability, regarding as a finite set of weights. We apply the simple scheme in Angelopoulos & Bates (2021, Section 1) to remove small weights in $\nu_v$, using a fraction of the validation set as the calibration set. The remaining weights are normalized to give $\nu_v'$. We evaluate by computing $\ell_v' = \|\nu_v' - \mu_v\|_1$ (against $\ell_v = \|\nu_v - \mu_v\|_1$).

From the results in Table 13, we see that conformal prediction indeed achieves the desired denoising effect for Texas, Cornell, and Wisconsin datasets. Consistently, for the difficult datasets Chameleon, Squirrel, and Actor, conformal prediction has minimal effect.

Table 13: The conformal prediction results.

| Method | Texas | Cornell | Wisconsin | Chameleon | Squirrel | Actor |
|---|---|---|---|---|---|---|
| GCN-I | 0.6874 | 0.7286 | 0.7204 | 0.7360 | 0.6240 | 0.9044 |
| GCN-I-C | 0.7281 | 0.7260 | 0.7423 | 0.9846 | 0.8718 | 0.9160 |
| GCN-I-AE | 0.6181 | 0.6482 | 0.5634 | 0.6934 | 0.5834 | 0.8877 |
| GCN-I-AE-C | 0.5886 | 0.5952 | 0.5328 | 0.9671 | 0.8834 | 0.9098 |

## F.3 More results on average $\ell_v$

In Table 14, we observe that the 1-layer GCN (GCN-I) indeed generally has the smallest average $\ell_v$, which agrees with what the theory predicts. This is because other eventual GCN models essentially change the graph structure and each $\nu_v$ is expected to be similar to the neighborhood label distribution in the new graph. As observed in Section 5, GCN-I-AE generally improves the metric of average $\ell_v$.

Table 14: The results on average $\ell_v$ for different models

| Method | Texas | Cornell | Wisconsin | Chameleon | Squirrel | Actor |
|---|---|---|---|---|---|---|
| GCN-I | 0.6874 | 0.7286 | 0.7204 | 0.7360 | 0.6240 | 0.9044 |
| GCN-I-AE | 0.6181 | 0.6482 | 0.5634 | 0.6934 | 0.5834 | 0.8877 |
| GCN-II | 0.7803 | 0.7435 | 0.7842 | 0.8899 | 0.7323 | 0.9550 |
| GCN-II-AE | 0.8182 | 0.7161 | 0.7218 | 0.8337 | 0.6961 | 0.9322 |
| GAT-I | 0.6686 | 0.6726 | 0.6975 | 1.0335 | 0.8966 | 0.9774 |
| GAT-I-AE | 0.6642 | 0.6598 | 0.6707 | 0.9090 | 0.8784 | 0.9156 |
| GAT-II | 0.7669 | 0.7652 | 0.7989 | 1.1305 | 1.0255 | 1.0305 |
| GAT-II-AE | 0.7229 | 0.7051 | 0.7235 | 1.0605 | 1.0213 | 0.8668 |
| ACM-GCN | 0.6702 | 0.6596 | 0.8062 | 1.0791 | 0.8308 | 0.9736 |
| ACM-GCN-AE | 0.6567 | 0.6788 | 0.7894 | 0.9760 | 0.7685 | 0.9214 |
| GraphCON | 0.7697 | 0.7890 | 0.8296 | 1.1419 | 1.0386 | 0.9804 |
| GraphCON-AE | 0.7337 | 0.7200 | 0.7879 | 1.1161 | 1.0297 | 0.9614 |
| CDE | 1.2348 | 1.1720 | 1.1424 | 1.2093 | 1.1140 | 0.8412 |
| CDE-AE | 1.2153 | 1.1676 | 1.1361 | 1.2078 | 1.1138 | 0.8374 |
| GloGNN | 1.6223 | 1.3085 | 1.4972 | 1.1801 | 0.9598 | 1.0368 |
| GloGNN-AE | 1.6397 | 1.3045 | 1.5049 | 1.1648 | 0.9561 | 1.0285 |

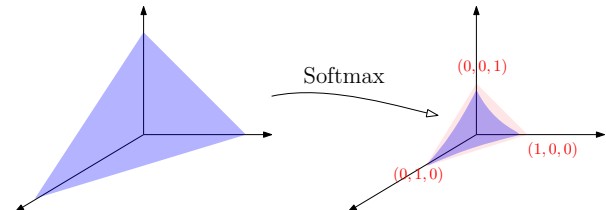

Figure 15: An illustration of the effect of the softmax function.

## G  LIMITATIONS

The framework provides insights into theoretical explanations for various GNN phenomena and enhances our understanding of the mechanisms and fundamental limitations of GNN models. Given a dataset, we aim to use the framework to rigorously assess its hardness quantitatively. Our theoretical and numerical results show that the quasi-isometric shape of the feature centroid simplex is useful in distinguishing between easier datasets (e.g., Texas) and more challenging ones (e.g., Actor). However, we have not yet proven that any numerical invariants associated with the shape can provide a guaranteed error (lower) bound, such as the Bayesian error, for any GNN models. This will be a subject of further investigation in future work.

Technically, the reliance of our framework on node features faces challenges when there are few useful features. For example, this occurs for many graph classification tasks (see Table 15), where the graph labels largely depend on the graph structure. For such a task, our proposed framework does not immediately give an insightful understanding of the dataset. A possible solution is to apply the framework in conjunction with position encoding of the nodes as features.

Table 15: Graph classification results(%). The base models are GIN (Xu et al., 2019) and PANDA (Choi et al., 2024). We apply the only relevant trick for classification: feature normalization (suffix:"-N"). We observe a slight improvement over the base model, which prompts further investigation is needed to understand how our framework might be adopted for graph classification.

| Method | MUTAG | Proteins | ENZYMES |
|---|---|---|---|
| GIN | 88.83±6.10 | 75.60±1.34 | 48.35±4.47 |
| GIN-N | 90.11±4.35 | 75.71±1.18 | 49.52±4.91 |
| PANDA | 88.49±4.90 | 74.46±2.22 | 46.33±4.73 |
| PANDA-N | 89.50±3.86 | 75.51±2.70 | 47.16±4.13 |

## H  MISCELLANEOUS

**The softmax function**   Given a feature vector, it is a common practice to obtain a probability distribution by applying the softmax function to the components. For the convenience of theoretical analysis, we have not discussed the effect of the softmax function. As the softmax function is not a linear transformation and is always nonzero, it will not match the vertices of a simplex $\Delta$ to that of the probability simplex $\Delta_c$. However, it is approximately so as illustrated in Fig. 15. Moreover, the softmax function is order-preserving (of the components), therefore, omitting the softmax function does not change prediction accuracy.

**The $\|\cdot\|_1$-evalutation metric**   In Section 5, we have proposed considering average $\ell_v$ as an evaluation metric. This is essentially an 1-norm. We choose this over the 2-norm because the 1-norm is the same as the Wasserstein metric on the discrete space $C$ of label classes (Villani, 2009; Ji et al., 2023a). Moreover, 1-norm reflects better sparsity under the condition that for any given node, most of its neighbors belong to 1 or 2 classes. However, the metric is used as a measure of the closeness between the predicted distribution $\nu_v$ and the ground-truth distribution $\mu_v$. Using 2-norm is also a reasonable choice.

**Normalized adjacency matrix**    In the early work Kipf & Welling (2017), it is already noted that the normalized version of the adjacency matrix should be used for convolution. On the other hand, Xu et al. (2019) claims that using the adjacency matrix as the aggregator is more expressive. For the node classification, our perspective suggests that feature aggregation is to reduce class feature variance. Therefore, the normalized adjacency matrix serves this purpose better.

