# OpenReview forum: "Rethinking Graph Neural Networks From A Geometric Perspective Of Node Features"
_ICLR.cc/2025/Conference — ICLR 2025 Poster_

### Official Review · Reviewer_p71D · 2024-10-28

**Soundness:** 3
**Presentation:** 3
**Contribution:** 3
**Rating:** 8
**Confidence:** 3

**Summary:**

The paper proposes to look at the node features from a simplex perspective. The main observations are a new view on oversmoothing and homo/heterophily. Further, some tricks (feature normalization and feature shuffling) are proposed to improve empirical results.

**Strengths:**

Node features and their proper handling has proven effective for a number of graph-learning tasks, so focusing on those instead of taking features as a second-class citizen after structure makes sense.
The main contribution of the paper is a theoretical model which allows to analyze GNNs wrt overfitting and homo/heterophily.
Experiments show that the presented tricks can indeed improve empirical performance.

**Weaknesses:**

The paper is unnecessarily hard to read. There is a lot of notation and the definitions and usage are sometimes far away (including non-standard notation).
The paper is also not really self-sufficient, there are very many mentions of the appendix promising e.g. experiments (even though some are in the paper the reference is to the appendix).
The probabilistic node feature model is not motivated. Since all results are based on this model, it would be good to at least argue in how far the assumptions are ``natural''. E.g. what does it mean for features to be convex? And does that appear in practice? (again, only a reference to the appendix...)
None of the theorems as an extensive description of what it means or why it is important. Usually its just the theorem and nothing else. E.g. 314 mentions an example that is no longer part of the main text.
The proposed feature normalization is as informal as it can get (which is surprising, given the formality in the rest of the paper). So its unclear whether each feature (across nodes) or each node's feature vector is normalized. The former is a standard trick and not new at all, so I guess its the latter.
The feature-shuffling trick is also not well-explained (but well-motivated). How would that be applied in a real-world dataset where e.g. 5-10% of the nodes are labeled?

Further small points:
- 226: G is undefined. (or rather, the quantifier is unclear)
- 220: please make clear that the model in the paper is not quite the original GCN but rather the version without sqrt(d). I believe that moving the formal definition of GCN in front of 224 would significantly improve readability.
- 406: how does this now differ from the spectral result? (looks pretty much like the same result)
- 472: the list of tricks that is applied did not make it into the main paper. I guess that should be changed.
- 527: please repeat what $\ell_v$ is about, this is not standard and most of all not just some standard accuracy.
- 537: "many" could be made specific.

**Questions:**

Overall, I believe that a long version of the paper (e.g. one for arxiv), with all examples, filler texts and some proofs re-inserted is a much better version of the paper. Currently, the setup is not well-motivated and the paper not self-contained. It also felt overly formal over large parts of the text.

I would thus like to encourage to work on the presentation.

---

> ### Author Response · Authors · 2024-11-21
> **Response to reviewer's comments Part 1**
>
> Thank you for the comments and suggestions.
>
> **Comment 1:** The paper is unnecessarily hard to read. There is a lot of notation and the definitions and usage are sometimes far away (including non-standard notation). The paper is also not really self-sufficient, there are very many mentions of the appendix promising e.g. experiments (even though some are in the paper the reference is to the appendix).
>
> **Response:** Thank you for the comments. This paper is mainly theoretical and it proposes a new perspective to analyze datasets and GNNs. Therefore, presenting technical results is essential to the paper.
>
> We hope that it is understandable that some material has to be delegated to the Appendices due to the page limit. In the main text, our focus is two-folded:
> - We present the main theoretical results while omitting the proofs.
> - We discuss their implications, including GNN phenomena and the proposed methods.
>
> Numerical evidence is needed to justify assumptions and claims. However, readers can omit these numerical results upon first reading, while verifying their validity later on if needed. We think this is a balanced approach given the page limit.
>
> We agree with the reviewer that the paper can benefit from re-organizing the materials, e.g., introducing all the new concepts together to give an overall picture as early as possible. We will make efforts towards this direction when revising the paper.
>
> **Comment 2:** The probabilistic node feature model is not motivated. Since all results are based on this model, it would be good to at least argue in how far the assumptions are ``natural''. E.g. what does it mean for features to be convex? And does that appear in practice? (again, only a reference to the appendix...)
>
> **Response:** Regarding Lemma 1, we have verified that the assumption holds for *all* the datasets in Sec. 5 Experiments. It also holds for most datasets used in the Appendices, with the "genius" dataset as the only exception. However, Table 11 shows that our proposed tricks work for the "genius" dataset as well.
>
> To explain the intuition, as we have discussed in Appendix B, there are two categories of features that make the result hold: (1) If each component of a feature is either $0$ or $1$ (e.g., Cora, Citeseer, etc.), then features of nodes are on a high-dimensional hypercube. In this case, the features are always in a convex position. (2) If each component of a feature is randomly chosen from $(0,1)$ (here we omit a few technicalities), then it is known that with high probability, the features are in a convex position provided the feature dimension is sufficiently large. To get a sense of "sufficiently large", if the feature dimension is $30$, then $10^7$ features are in a convex position with probability $\geq 99\\%$ [a]. For most ML applications, feature dimensions are likely to be $>30$ (except for some special cases), which makes Lemma 1 widely applicable.
>
> We agree with the reviewer that our paper is proposing a feature-centric theoretical framework to analyze GNNs. Therefore, assumptions on features are needed as a starting point. As we have discussed above, we believe they are not particularly stringent.
>
> [a]: A. Gorban, et. al. Stochastic separation theorems: How geometry may help to correct AI errors, Notice AMS, 70(1):25-33, 2022.
>
> **Comment 3:** None of the theorems as an extensive description of what it means or why it is important. Usually its just the theorem and nothing else. E.g. 314 mentions an example that is no longer part of the main text.
>
> **Response:** We have realized this potential problem of being technical and tried a few approaches to convey our main ideas better. For example, we have provided numerous numerical results to support the theoretical results. Immediately after each theorem, we have dedicated a paragraph (highlighted with "{\bf Discussion}") to discuss the intuitions of the theorem and its implications. In the discussion, we have connected the theory to the proposed tricks whenever possible.
>
> Regarding the example in 314, it is a numerical experiment that verifies the statement "... the GCN layer generates a probability distribution that is almost the same as the neighborhood class label distribution of $v$, with high probability... ". The statement, on the other hand, is an implication of Theorem 2.

---

> > ### Comment · Reviewer_p71D · 2024-11-21
> >
> > Wrt. unnecessarily hard to read: I fully understand that this is a theoretical paper and thus clear notation is of utmost importance. But already the very first formal sentence (121-126) contains both $m$ (the feature dimension which surprisingly is not called $d$) and $m_1,m_2$ (just some constants) which seem to be completely unrelated. Then there is a reference to the appendix where naming two datasets would also have worked for the purpose of the text. The next line again refers to features (I would have preferred calling them "feature vectors" and one has to search for the notation (for me the distinction between individual features ("feature component values") in [0,1] and "features" $x^i \in [0,1]^m$ could have been more clear - it took me way too long to figure out where those features $x^1,\dots,x^n$ (l. 130) were referring to). Next line you write for each index $i$ where I would have expected "For each node $v_i$ with $i = 1,\dots n$". Then one has again to search for $\gamma$ only to find out that this is probably undefined and "over C" is missing in that line (only $\gamma_c$ has been used before). Next comes the lemma, but without the explanation from the rebuttal.
> >
> > I know all of these are super minor details and given enough time, it is no problem to figure out exactly what is meant, but in my opinion this section 2.1 just gives a bad start into the formal part of the paper where minimal changes would have made it much easier to read for me.
> > I would like to emphasize that it gets easier to read further into the paper which is good. I also liked the numerical experiments, since they clearly emphasize the main points of the paper.
> >
> > Comment 2: adding the paragraph from the rebuttal below Lemma 1 would massively improve the paper since it makes clear that the overall model "makes sense" and is not completely arbitrary. (especially since I am not the only one asking about it)
> >
> > Comment 3: thanks for pointing out that the main theorems indeed each come with a discussion paragraph (just the lemmas often do not). Wrt 314, I would have expected the sentence to read like "is in App E.3, showing ..." (i.e. being more specific than a simple "verifying" at the beginning of the sentence). At least to me it is not obvious from the description what one would find in App E.3, maybe it is possible to reformulate the sentence? The "thought experiment" mentioned there is also rather vague.
> >
> > Overall, I still like the paper's approach and results and mainly struggle with the presentation. E.g. If the part after Lemma 1 is added, I would no longer consider the setup weakly-motivated. And by self-contained I mostly meant that it is often unclear what exactly is hidden in the appendix with the references being plentiful but undetailed.
> >
> > Even though not part of my review, I really liked that you additionally checked to use your simplex approach for oversmoothing which might be a nice and principled way of measuring its effects. Formally comparing that to other approaches might be an interesting direction, but is possibly beyond the scope of this paper.

---

> > > ### Author Response · Authors · 2024-11-22
> > >
> > > Thank you very much for your helpful comments and suggestions. We have submitted a revised version of the paper incorporating comments from all the reviewers. Regarding your concerns, we have reorganized Section 2.1 by removing unnecessary technical details and providing more intuitions (in view of your Comment 2). We also have reorganized Section 2 and Section 3 so that all the base geometric models are introduced together (towards the end of Section 2.2), hoping the overall structure is more coherent. The other changes are made accordingly in the revision.
> > >
> > > We shall keep improving the paper. Thank you for your precious time in reviewing our work.

---

> > > > ### Comment · Reviewer_p71D · 2024-11-22
> > > >
> > > > Thanks a lot for restructuring! In my opinion the paper is much easier to read this way! I have thus raised my score as my concerns have been solved.

---

> > > > > ### Author Response · Authors · 2024-11-22
> > > > >
> > > > > Thank you very much. We believe that our paper has benefited greatly from your suggestions.

---

> ### Author Response · Authors · 2024-11-21
> **Response to reviewer's comments Part 2**
>
> **Comment 4:** The proposed feature normalization is as informal as it can get (which is surprising, given the formality in the rest of the paper). So it's unclear whether each feature (across nodes) or each node's feature vector is normalized. The former is a standard trick and not new at all, so I guess it's the latter.
>
> **Response:** The reviewer is correct. It is the latter approach being adopted. We will clarify this when revising the paper. As we have explained in the paper, the trick aims to reshape the simplex $\Delta_g$, when it resembles a degenerate simplex (as discussed in Section 3.3). Datasets such as Chameleon, Squirrel, and Actor satisfy such a property. The trick is demonstrated to work for these datasets.
>
> **Comment 5:** The feature-shuffling trick is also not well-explained (but well-motivated). How would that be applied in a real-world dataset where e.g. $5-10\%$ of the nodes are labeled?
>
> **Response:** Feature reshuffling discussed in the paper is not a trick, but rather a phenomenon. The theory predicts that there are two types of feature reshuffling phenomenon depending on whether the simplex $\Delta_g$ is regular or degenerate, which are verified by numerical results in Table 1. The phenomenon supports our theory. Moreover, it shows the intrinsic difficulty of some datasets (e.g., Actor), and explains why most GNN models do not perform well for these datasets.
>
> **Comment 6:** Further small points.
>
> 226: G is undefined. (or rather, the quantifier is unclear)
>
> 220: please make clear that the model in the paper is not quite the original GCN but rather the version without sqrt(d). I believe that moving the formal definition of GCN in front of 224 would significantly improve readability.
>
> 406: how does this now differ from the spectral result? (looks pretty much like the same result)
>
> 472: the list of tricks that is applied did not make it into the main paper. I guess that should be changed.
>
> 527: please repeat what $\ell_v$ is about, this is not standard and most of all not just some standard accuracy.
>
> 537: "many" could be made specific.
>
> **Response:** Thank you for these comments and suggestions. We will address them while revising the paper.
> - 226: $G$ is the original graph of the dataset, which we will clarify.
> - 220: We will revise as suggested.
> - 406: Our approach and the spectral approach are two different perspectives of the same oversmooth phenomenon. Our explanation is more explicit on the dynamics of the simplex $\Delta_g$ and thus can be easily visualized. The stochastic matrix describing the dynamic is different from that used in the spectral approach.
> - 472: The tricks are (1) and (2) described at the beginning of the subsection "Simple tricks" in Sec. 5 Experiments.
> - 527: We will repeat the definition of $\ell_v$ in the table caption as suggested.
> - 537: We will explicitly cite the phenomenons in Sec. 5 as suggested.
>
> **Questions:** Overall, I believe that a long version of the paper (e.g. one for arxiv), with all examples, filler texts and some proofs re-inserted is a much better version of the paper. Currently, the setup is not well-motivated and the paper not self-contained. It also felt overly formal over large parts of the text.
>
> I would thus like to encourage to work on the presentation.
>
> **Response:** We shall follow your suggestion and prepare an arXiv version of the paper with the appendices inserted into the main text.

---

### Official Review · Reviewer_ZoMg · 2024-10-28

**Soundness:** 3
**Presentation:** 2
**Contribution:** 2
**Rating:** 6
**Confidence:** 5

**Summary:**

This paper introduces a *feature-centric approach* to analyzing GNNs by examining the node embeddings for each class in feature space  rather than graph topology. The authors define a *feature centroid simplex* and leverage coarse geometry to explore GNN behaviors like heterophily and oversmoothing, offering new insights and simple techniques for improving node classification performance. They make extensive experiments to validate their analysis

**Strengths:**

1. The paper introduces an innovative approach to studying GNNs by focusing on the clustering behavior of node embeddings for each class in feature space.
2. By defining a *feature centroid simplex* as a (K-1)-simplex using the centers of these clusters, where K is the number of classes, the authors provide a geometric representation in $\mathbb{R}^m$ that effectively captures class relationships.
3. They use this geometric perspective to explain key GNN challenges, such as homophily/heterophily and oversmoothing, in terms of the feature centroid simplex.
4. The mapping of the probability simplex to the *feature centroid simplex* is particularly insightful, as it reveals additional learning opportunities within the feature space.

**Weaknesses:**

1. **Presentation Quality**: The paper’s main weakness lies in the presentation, which does not effectively communicate its promising ideas. Although grounding concepts in mathematical rigor is essential, the technicalities in this paper obscure the overall clarity and may hinder comprehension.

2. **Mathematical Statements**: The mathematical claims in Sections 2 and 3 lack interpretability. The space of graphs with feature vectors is vast and diverse, requiring more restrictive assumptions or narrowing to particular subclasses for these statements to hold meaning. For instance, Lemma 1 is difficult to justify unless the feature dimension m is very large and comparable to the number of nodes n. Similarly, Lemma 2 could benefit from a focus on the geometric simplex rather than the ambiguous term $e_c$, which complicates the exposition. Such imprecise statements make the argument obscure and challenging to follow.

3. **Distinction of Homophily and Heterophily**: The idea to distinguish homophily and heterophily through feature centroid differences or clustering behaviors of node neighborhoods is promising. However, adding a geometric quantification—such as using the diameter, the volume of the simplex, or embedding of node neighborhoods with normalization—would strengthen the approach and add greater empirical rigor.

4. **Oversmoothing Analysis**: Analyzing changes in the feature centroid simplex under oversmoothing is an excellent idea, with the potential to develop a metric for measuring oversmoothing. To achieve this, the paper would benefit from more precise definitions of the geometric quantities involved, along with a demonstration of changes in these quantities across epochs. For example, seeing the changes in the volume of feature centroid simplex under each epoch for oversmoothing would be very interesting.

5. **Incremental Improvements**: Although the simple tricks suggested are useful, they offer only incremental improvements. A more substantial contribution, such as a more comprehensive approach that integrates the meaningful information (e.g., some geometric quantities) obtained from feature centroid simplex approach to ML pipelines, would significantly strengthen the impact of this work.

**Questions:**

See weaknesses.

---

> ### Author Response · Authors · 2024-11-21
> **Response to reviewer's comments Part 1**
>
> Thank you for the comments and suggestions.
>
> **Comment 1:** Presentation Quality: The paper’s main weakness lies in the presentation, which does not effectively communicate its promising ideas. Although grounding concepts in mathematical rigor is essential, the technicalities in this paper obscure the overall clarity and may hinder comprehension.
>
> **Response:** Thank you for the comments. We propose a new theoretical perspective to analyze datasets and GNNs. Based on the theoretical insights, We also propose tricks to improve GNN performance. To better convey our main ideas, we have provided numerous numerical results to support the theoretical results. After each theorem, we provide a thorough discussion of the intuitions and implications.
>
> We agree with the reviewer that the paper can benefit from re-organizing the materials, e.g., introducing all the new concepts together to give an overall picture as early as possible. We will make efforts towards this direction when revising the paper.
>
> **Comment 2:** Mathematical Statements: The mathematical claims in Sections 2 and 3 lack interpretability. The space of graphs with feature vectors is vast and diverse, requiring more restrictive assumptions or narrowing to particular subclasses for these statements to hold meaning. For instance, Lemma 1 is difficult to justify unless the feature dimension m is very large and comparable to the number of nodes n. Similarly, Lemma 2 could benefit from a focus on the geometric simplex rather than the ambiguous term $e_c$, which complicates the exposition. Such imprecise statements make the argument obscure and challenging to follow.
>
> **Response:** We agree with the reviewer that is essential to provide intuitions to the main theoretical results. Immediately after each theorem, we have dedicated a paragraph (highlighted with "**Discussion**") to discuss the intuitions of the theorem and its implications. In the discussion, we have connected the theory to the proposed tricks whenever possible.
>
> Regarding Lemma 1, we have verified that the conclusion holds for *all* the datasets in Sec. 5 Experiments. It also holds for most datasets used in the Appendices, with the "genius" dataset as the only exception. However, Table 11 shows that our proposed tricks work for "genius" as well.
>
> As we have explained in the paper, the probabilistic centroids $\\{e_c\mid c\in C\\}$ are more convenient for theoretical analysis, as tools such as concentration inequalities are available. Therefore, Lemma 2, as a theoretical result, is stated in terms of the probabilistic centroids. The geometric centroids can be readily computed from samples, and hence they are useful as proxies in practice.
>
> **Comment 3:** Distinction of Homophily and Heterophily: The idea to distinguish homophily and heterophily through feature centroid differences or clustering behaviors of node neighborhoods is promising. However, adding a geometric quantification—such as using the diameter, the volume of the simplex, or embedding of node neighborhoods with normalization—would strengthen the approach and add greater empirical rigor.
>
> **Response:** Recall that a dataset is heterophilic if there are many pairs of nodes belonging to different classes that are connected by an edge. Being homophily or heterophily is thus in fact a feature-agnostic property that depends only on the graph and labels. Hence, homophily and heterophily are not directly related to geometric properties of the simplex $\Delta_g$.
>
> However, we have theoretically obtained the likely position (in Theorem 1) of the aggregated feature of each node in the simplex $\Delta_e$. It is the mixed centroid given in (4). Therefore, such a nodewise positional information allows us to devise approaches to handle heterophily, while not global geometric information of $\Delta_e$ itself.

---

> ### Author Response · Authors · 2024-11-21
> **Response to reviewer's comments Part 2**
>
> **Comment 4:** Oversmoothing Analysis: Analyzing changes in the feature centroid simplex under oversmoothing is an excellent idea, with the potential to develop a metric for measuring oversmoothing. To achieve this, the paper would benefit from more precise definitions of the geometric quantities involved, along with a demonstration of changes in these quantities across epochs. For example, seeing the changes in the volume of feature centroid simplex under each epoch for oversmoothing would be very interesting.
>
> **Response:** Thank you for the insightful suggestion. We have modified your idea and designed the following synthetic experiment: We train a deep GCN and study the volume $\text{Vol}_k(\Delta_g)$ of the resulting $\Delta_g$ after the $k$-th layer, expecting to see it diminishing. As a technical issue, the Euclidean volume $\text{Vol}_k(\Delta_g)$ is always $0$ as the feature dimension is larger than the number of classes, therefore, we compute the average volume of different projections of $\Delta_g$ to smaller dimensional subspaces of the feature space.
>
> We have studied Cora and Texas datasets as representatives of homophilic and heterophilic datasets. For the Cora dataset, after $4$ layers, the (average) volume is reduced to $\approx 1.7\\%$ of the initial (average) volume. For the Texas dataset, after $7$ layers, the (average) volume is reduced to $\approx 2.6\\%$ of the initial (average) volume. The observations support the theory.
>
> **Comment 5:** Incremental Improvements: Although the simple tricks suggested are useful, they offer only incremental improvements. A more substantial contribution, such as a more comprehensive approach that integrates the meaningful information (e.g., some geometric quantities) obtained from the feature centroid simplex approach to ML pipelines, would significantly strengthen the impact of this work.
>
> **Response:** Thank you for the comments. The emphasis of the paper is to propose a new perspective to analyze datasets and GNNs, rather than developing a full-fledged GNN model. As explained in the introduction, we want to fundamentally understand how hard a dataset is by using the proposed framework. We see that from the numerical results, the more difficult datasets (in the sense that most approaches have relatively worse results) such as Squirrel and Actor have less regular feature simplex shape, which agrees with what the theory predicts.
>
> The proposed tricks are devised based on the theory. They are very simple to implement with low complexity. From the results, the performance improvement is consistent over most datasets and most base models (e.g., we have discussed in the paper that 34/48 improvements in Table 2 are statistically significant). Moreover, we can provide insights into why it is hard to get better results for certain datasets and provide an alternative solution such as applying the normalization trick.
>
> Therefore, the performance of the proposed tricks supports the validity of the theoretical framework devised to analyze datasets. However, the tricks or the model performance are not the main focus of the paper.

---

> ### Comment · Reviewer_ZoMg · 2024-11-22
>
> Thank you very much for detailed responses, and volume experiments. The revision looks much better. My concerns are mainly addressed. I am raising my score.

---

> > ### Author Response · Authors · 2024-11-22
> >
> > We sincerely thank you for taking the time to review our revised manuscript and for your thoughtful feedback. We are delighted to hear that the changes have addressed your concerns. We greatly appreciate your recognition of our efforts and are truly grateful for your decision to raise the score. Your constructive comments have significantly contributed to improving the quality of our work, and we deeply value your support.

---

### Official Review · Reviewer_NSvv · 2024-11-04

**Soundness:** 3
**Presentation:** 3
**Contribution:** 2
**Rating:** 6
**Confidence:** 3

**Summary:**

The paper "Rethinking Graph Neural Networks from a Geometric Perspective of Node Features" proposes a new framework for understanding Graph Neural Networks (GNNs) by focusing on node features rather than solely on graph topology. The authors introduce the "feature centroid simplex," a geometric structure representing the distribution of node features across classes, as a tool to explore phenomena like heterophily (when nodes connect across classes), oversmoothing, and feature reshuffling in GNNs. This feature-centric perspective addresses limitations of traditional topology-based approaches, especially for datasets where graph structure alone does not fully capture data complexity. The concept of the "feature centroid simplex," a high-dimensional simplex formed by the centroids of node features within each class introduced by the authors, is presented as a geometric model that enables a deeper understanding of key GNN phenomena, such as heterophily, oversmoothing, and feature aggregation. By studying the convex shapes of these simplexes and their proximities, the paper explains the behavior of GNNs across different graph structures, especially on challenging tasks like node classification. It also suggests simple, graph-independent tricks to improve GNN performance, such as early stopping and feature normalization, particularly on heterophilic datasets where standard GNNs often struggle. The paper works to provide a unified geometric framework to enhance theoretical understanding and practical efficacy in GNN applications​

**Strengths:**

The paper’s novelty lies in its shift from topology-based to feature-centric analysis in understanding GNNs, introducing the concept of a "feature centroid simplex" formed by class-based feature centroids, which allows GNN behavior to be studied within feature spaces rather than solely relying on graph structure. By applying coarse geometry to examine simplex shapes, the paper offers new insights into common GNN challenges, including heterophily, oversmoothing, and feature variance. This graph-agnostic, feature-driven approach provides a unique way to assess dataset-specific difficulties, such as classification "hardness," focusing on intrinsic feature properties rather than external graph characteristics. The primary contributions of the paper include developing a theoretical framework based on the feature centroid simplex, which offers new perspectives on GNN phenomena. It also provides practical, graph-independent tricks like early stopping and adding edges between nodes of the same class to improve GNN performance in difficult settings. These insights are supported by empirical validation on heterophilic datasets such as Actor, Chameleon, and Squirrel, where the feature-centric approach and proposed techniques demonstrate substantial improvements over traditional GNN models.

The paper provides a clear, succinct abstract that outlines its goal to rethink Graph Neural Networks (GNNs) from a feature-centric, geometric perspective. The structure flows logically from a foundational background to theoretical concepts like the feature centroid simplex and quasiisometry, and then to practical applications for GNN improvement. Each section builds upon the previous, and a notation table in the appendix enhances clarity.

The paper demonstrates theoretical rigor with a strong foundation in geometric and probabilistic models, supported by detailed proofs in the appendices

Visuals, including feature centroid simplexes, effectively clarify complex ideas, and experiments are presented with comprehensive tables and discussions on framework effectiveness across datasets.

Appendices are well-used to include supplementary proofs, notation, and experiments, keeping the main content focused while providing technical depth for interested readers.  Overall, the structure, visual aids, and appendices provide a strong foundation for understanding the paper’s contributions.

The provided illustration were found to be well done and very informative.

**Weaknesses:**

The level of novelty and degree of contributions in the work presented were not fully convincing and could perhaps be made clearer by making explicitly the degree to which the proposed work contributes as well as a more explicit discussion on where the proposed work has limitations. For example, the feature-centric approach, while novel, may have limited applicability in cases where feature information alone is insufficient, or where the graph structure itself is crucial, such as in social networks or molecular graphs where node connections are integral to understanding relationships. The framework and tricks presented are mainly beneficial for heterophilic datasets; however, for homophilic graphs, where existing GNN models already perform well, the proposed tricks may not offer significant improvements. Additionally, the tricks suggested, such as early stopping and feature normalization, are simple yet not inherently innovative, as these techniques are standard practices in machine learning for managing overfitting and feature scaling; the novelty lies more in their application to GNNs than in the techniques themselves.

Another limitation is the paper’s reliance on certain assumptions about feature distribution and convexity, which may not hold across all types of graph data, potentially limiting the framework’s generalizability, especially when features are not easily separable or node classes lack well-defined centroids.

In terms of novel contributions, the paper's innovation is somewhat limited. Although the feature-centric perspective provides a new approach, it builds on well-known GNN challenges such as oversmoothing and heterophily without fundamentally altering GNN structures or proposing new architectures. Thus, the framework offers insights rather than resolutions to these issues. Furthermore, the recommended tricks, while practical, are widely used machine learning techniques, and applying them to GNNs does not significantly advance the field. The empirical comparisons primarily focus on baseline models and lack a thorough analysis of how more advanced GNNs, with adaptive architectures or attention mechanisms, might perform with these tricks, which constrains the paper’s insights into how its geometric perspective could be leveraged in state-of-the-art models.

The paper would benefit from an experimental evaluation with datasets with low label informativeness to assess the quality of their feature centric approach such as those offered by recent works such as “A critical look at the evaluation of GNNs under heterophily: Are we really making progress?” by Platonov et. al. A more in depth comparative analysis to other similar methods which perform topological or geometric analysis of the feature space to implore learning model performance such as persistence landscapes or persistence images by Adams et. al.and other methods such as those discussed in Architectures of Topological Deep Learning: A Survey of Message-Passing Topological Neural Networks by Papillon et. al. should be experimentally compared to and definitely discussed in the background and related works section. The authors should also consider comparing to other geometry based approaches as presented in “Feature Expansion for Graph Neural Networks” by Sun et. al. or “A Survey of Geometric Graph Neural Networks: Data Structures, Models and Applications” by  Han, Cen, and Wu et. al. however I have not extensively read these last two works.

On reading the paper, I was not convinced of the validity and generalized applicability of the approach due to the assumptions that were required in order to provide proof of necessary lemmas and theorems as presented and needed to support the paper's claims. A more detailed analysis of the bounds in the number of sample points is needed in order for the probabilistic claims to be met such as convex positioning or the number of samples needed to justifiably use the geometric centroid of a class.

The related works discussed in the introduction and background should be expanded as well as more explicit discussion of the proposed paper in the context of contemporary literature. An more explicit discussion with the novelty of the current work could be made more persuasive. A more explicit limitations section and discussion would also be appreciated.

Definitions of concepts used and background information is at times distributed throughout the paper rather than presented at first in background and related work. Background and foundational concepts should be presented earlier and consolidated.

**Questions:**

Can the assumptions made to provide proofs to lemmas and theorems be listed and justified as reasonable?

Can the limitations caused by the assumptions made be discussed?

Can the limitations of the approach be made more explicit, and despite those limitations, the novely and contributions of the proposed approach be justified?

---

> ### Author Response · Authors · 2024-11-21
> **Response to reviewer's comments Part 1**
>
> Thank you for the comments and suggestions.
>
> **Comment 1:** The level of novelty and degree of contributions in the work presented were not fully convincing and could perhaps be made clearer by making explicitly the degree to which the proposed work contributes as well as a more explicit discussion on where the proposed work has limitations. For example, the feature-centric approach, while novel, may have limited applicability in cases where feature information alone is insufficient, or where the graph structure itself is crucial, such as in social networks or molecular graphs where node connections are integral to understanding relationships. The framework and tricks presented are mainly beneficial for heterophilic datasets; however, for homophilic graphs, where existing GNN models already perform well, the proposed tricks may not offer significant improvements. Additionally, the tricks suggested, such as early stopping and feature normalization, are simple yet not inherently innovative, as these techniques are standard practices in machine learning for managing overfitting and feature scaling; the novelty lies more in their application to GNNs than in the techniques themselves.
>
> **Response:** Thank you for the useful suggestions. We agree with the reviewer that a section on the limitations of our work can be beneficial to the readers, which is what we will include in the revision:
>
> "**Limitations**: The framework allows us to have a glimpse into theoretical explanations of many GNN phenomena, and to enhance our understanding of the mechanisms and fundamental limitations of GNN models. Given a dataset, we also aspire to use the framework to rigorously identify the hardness. From our theoretical and numerical results, the quasi-isometric shape of the feature centroid simplex demonstrates its usefulness in distinguishing easy datasets (e.g., Taxes) from hard ones (e.g., Actor). However, we are not able to prove that any numerical invariants associated with the shape can give a guaranteed error (lower) bound, i.e., the Bayesian error, for any GNN models. We will investigate further in future works.
>
> Technically, the reliance of the framework on node features faces challenges when there are few useful features. For example, this occurs for many graph classification tasks, where the label is solely determined by the graph structure. For such a task, our proposed framework does not immediately give an insightful understanding of the dataset. A possible solution is to apply the framework in conjunction with position encoding of the nodes as features. This is also a viable extension of our current work. "
>
> Regarding the novelty of the proposed tricks, please refer to Comment 3 below for a detailed discussion.
>
> **Comment 2:** Another limitation is the paper’s reliance on certain assumptions about feature distribution and convexity, which may not hold across all types of graph data, potentially limiting the framework’s generalizability, especially when features are not easily separable or node classes lack well-defined centroids.
>
> **Response:** We understand the reviewer's concern. We have verified that Lemma 1 holds for *all* the datasets in Sec. 5 Experiments, as well as standard homophilic datasets such as Cora, Citeseer, PubMed.
>
> The assumptions are not as unrealistic as one may expect, as we discuss in Appendix B. There are two categories of features that fit the assumptions: (1) If each component of a feature is either $0$ or $1$ (e.g., Cora, Citeseer, etc.), then features of nodes are on a high-dimensional hypercube. In this case, the features are always in a convex position. (2) If each component of a feature is randomly chosen from $(0,1)$ (here we omit a few technicalities), then it is known that with high probability, the features are in a convex position provided the feature dimension is sufficiently large. To get a sense of "sufficiently large", if the feature dimension is $30$, then $10^7$ features are in a convex position with probability $\geq 99\\%$ [a]. For most ML applications, feature dimensions are likely to be $>30$ (except for some special cases), which makes Lemma 1 widely applicable.
>
> We agree with the reviewer that our paper is proposing a feature-centric theoretical framework to analyze GNNs. Therefore, assumptions on features are needed as a starting point. As we have discussed above, we believe they are not particularly stringent.
>
> [a]: A. Gorban, et. al. Stochastic separation theorems: How geometry may help to correct AI errors, Notice AMS, 70(1):25-33, 2022.

---

> ### Author Response · Authors · 2024-11-21
> **Response to reviewer's comments Part 2**
>
> **Comment 3:** In terms of novel contributions, the paper's innovation is somewhat limited. Although the feature-centric perspective provides a new approach, it builds on well-known GNN challenges such as oversmoothing and heterophily without fundamentally altering GNN structures or proposing new architectures. Thus, the framework offers insights rather than resolutions to these issues. Furthermore, the recommended tricks, while practical, are widely used machine learning techniques, and applying them to GNNs does not significantly advance the field. The empirical comparisons primarily focus on baseline models and lack a thorough analysis of how more advanced GNNs, with adaptive architectures or attention mechanisms, might perform with these tricks, which constrains the paper’s insights into how its geometric perspective could be leveraged in state-of-the-art models.
>
> **Response:** Thank you for the comments. It is indeed true that the emphasis of the paper is to propose a new perspective to analyze datasets and GNNs, rather than developing a full-fledged GNN model. As explained in the introduction, we want to fundamentally understand how hard a dataset is using the proposed framework. In this respect, the paper has achieved progress. We see that from the numerical results, the more difficult datasets (in the sense that most approaches have relatively worse results) such as Squirrel and Actor have less regular feature simplex shape, which agrees with what the theory predicts.
>
> The proposed tricks belong to the following generic categories of machine learning techniques: *rewiring*, *early stopping*, and *feature normalization*. However, different research works propose different realizations of the above categories of machine learning techniques. The theoretical motivation and the explicit procedure differentiate approaches belonging to the same category. In Appendix C, we provide a detailed discussion about how our tricks are different from existing ones belonging to the same category.

---

> ### Author Response · Authors · 2024-11-21
> **Response to reviewer's comments Part 3**
>
> **Comment 4:** The paper would benefit from an experimental evaluation with datasets with low label informativeness to assess the quality of their feature centric approach such as those offered by recent works such as “A critical look at the evaluation of GNNs under heterophily: Are we really making progress?” by Platonov et. al. A more in depth comparative analysis to other similar methods which perform topological or geometric analysis of the feature space to implore learning model performance such as persistence landscapes or persistence images by Adams et. al.and other methods such as those discussed in Architectures of Topological Deep Learning: A Survey of Message-Passing Topological Neural Networks by Papillon et. al. should be experimentally compared to and definitely discussed in the background and related works section. The authors should also consider comparing to other geometry based approaches as presented in “Feature Expansion for Graph Neural Networks” by Sun et. al. or “A Survey of Geometric Graph Neural Networks: Data Structures, Models and Applications” by Han, Cen, and Wu et. al. however I have not extensively read these last two works.
>
> **Response:** Thank you for the insightful suggestion. As our paper is about node features, we believe it is more relevant to perform numerical study on low "feature" (instead of "label") informativeness. We consider the following setup: we randomly remove a fraction of features and consider the cases with $1/2, 1/4, 1/8$ features remaining for each node. We choose the overall best model ACM-GCN (see Table 2 in the paper) as the base model and test the performance of the proposed tricks. The results are shown in the table below and we see that the tricks generally improve the performance.
>
> **Table:** Node classification results (%).
>
> | Feature Dimension | Full           | 1/2           | 1/4           | 1/8         |
> |-------------------|----------------|---------------|---------------|---------------|
> | **Chameleon**     |                |               |               |               |
> | ACM-GCN           | 71.29±8.06     | 61.95±7.13    | 52.70±5.36    | 44.17±3.87    |
> | ACM-GCN-AEN       | 75.22±9.99     | 63.53±7.37    | 56.09±5.36    | 46.07±4.92    |
> | **Squirrel**      |                |               |               |               |
> | ACM-GCN           | 55.07±8.96     | 49.67±7.43    | 42.10±5.08    | 36.27±2.72    |
> | ACM-GCN-AEN       | 59.47±11.41    | 50.49±7.81    | 44.10±5.93    | 37.67±2.38    |
> | **Actor**         |                |               |               |               |
> | ACM-GCN           | 37.28±2.76     | 33.74±2.34    | 31.32±1.67    | 26.89±0.97    |
> | ACM-GCN-AEN       | 47.62±6.38     | 35.09±1.91    | 32.33±1.53    | 29.23±0.85    |
>
> It is indeed true that there are recent advancements in topological and geometric GNNS. They are different from our theme as they focus on more complicated topological and geometric structures of the networks, while we are studying the geometric properties of the features. It is worthwhile to include the following comparison at the conceptual level in Appendix C:
>
> "Recent works have used more complicated topological and geometric tools to model node correlations or to capture hidden structural network information. For example, [a] applies topological data analysis to retrieve global topological information such as the number of cycles to enhance GNN expressiveness. The survey [b] contains a comprehensive overview of models when graphs are replaced with more complicated and expressive structures such as cellular complexes and hypergraphs. On the other hand, [c] on geometric graph neural networks summarizes approaches given additional geometric information such as explicit locations of nodes in a 3D space. They put emphasis on geometric invariance and they are particularly useful for bio-chemical datasets. Unlike these works, our theme is on the geometry of the features instead of the network structure. Hence, though geometric and topological tools are employed, the subjects of study are different. "
>
> [a] M. Horn, et. al. Topological graph neural networks. ICLR, 2022.
>
> [b] M. Papillon, et. at. Architectures of Topological Deep Learning: A Survey of Message-Passing Topological Neural Networks. arXiv 2304.10031, 2024.
>
> [c] J. Han, et. al. A Survey of Geometric Graph Neural Networks: Data Structures, Models and Applications. 	arXiv:2403.00485, 2024.

---

> ### Author Response · Authors · 2024-11-21
> **Response to reviewer's comments Part 4**
>
> **Comment 5:** On reading the paper, I was not convinced of the validity and generalized applicability of the approach due to the assumptions that were required in order to provide proof of necessary lemmas and theorems as presented and needed to support the paper's claims. A more detailed analysis of the bounds in the number of sample points is needed in order for the probabilistic claims to be met such as convex positioning or the number of samples needed to justifiably use the geometric centroid of a class.
>
> **Response:** Regarding the convex positioning of node features, a detailed discussion is presented in our response to Comment 2. In summary, Lemma 1 holds for all except one dataset ("genius" used in the appendix). As we usually do not gain access to the probabilistic centroids, it is customary to analyze their asymptotic relation with the geometric centroids, which is given in Lemma 2.
>
> For the theorems, the only assumption is on the quasi-isometric shape of the simplexes associated with the dataset. As the theme of the paper is about how these shapes affect the "hardness" of datasets, the assumptions in the results are minimal in this respect.
>
> We do not make or need assumptions on the number of sample points. This is because such information is already contained in the mixed centroid defined in (4). It incorporates the number of nodes $|N_v|$ in a neighborhood $N_v$ of a node $v$. In our theoretical bound (Theorem 1(b)), sizes of subsets of $N_v$ are used explicitly. Numerical evidence of the validity of the results are given in the appendix (see the "discussion" after each theorem).
>
> **Comment 6:** The related works discussed in the introduction and background should be expanded as well as more explicit discussion of the proposed paper in the context of contemporary literature. An more explicit discussion with the novelty of the current work could be made more persuasive. A more explicit limitations section and discussion would also be appreciated.
>
> **Response:** The paper has Appendix C dedicated to related works. The focus is to explain how our proposed methods are different from those in the literature belonging to the same category. We have put emphasis on how our theoretical analysis leads to the proposed approaches. We will expand this section according to our response to Comment 4.
>
> **Comment 7:** Definitions of concepts used and background information is at times distributed throughout the paper rather than presented at first in background and related work. Background and foundational concepts should be presented earlier and consolidated.
>
> **Response:** We will follow the reviewer's suggestion to re-organize the content such that Definition 6 is introduced in Sec. 2 so that Sec. 2 contains all the foundational concepts.
>
> **Questions:** Can the assumptions made to provide proofs to lemmas and theorems be listed and justified as reasonable?
>
> Can the limitations caused by the assumptions made be discussed?
>
> Can the limitations of the approach be made more explicit, and despite those limitations, the novely and contributions of the proposed approach be justified?
>
> **Response:** Please refer to our responses to the comments (weaknesses) for details.

---

> ### Author Response · Authors · 2024-11-25
>
> Dear Reviewer NSvv,
>
> As the discussion period nears its end, we hope that we have effectively addressed and resolved your concerns. Your feedback on our rebuttal responses would be greatly appreciated, and we are more than willing to provide further clarification on any remaining issues. Thank you for your time and consideration.

---

> > ### Comment · Reviewer_NSvv · 2024-11-26
> >
> > With the added figures, well-articulated justifications, clarifying points to my own confusion, and added tables both here and in other responses, I have changed my rating to a 6 and find it suitable for acceptance.

---

> > > ### Author Response · Authors · 2024-11-26
> > >
> > > We sincerely thank you for taking the time to review our revised manuscript and for your thoughtful feedback. Your constructive comments have significantly contributed to improving the quality of our work, and we deeply value your support.

---

### Official Review · Reviewer_kubS · 2024-11-09

**Soundness:** 3
**Presentation:** 2
**Contribution:** 3
**Rating:** 8
**Confidence:** 3

**Summary:**

The paper proposes a novel perspective for studying datasets used in node classification, which focuses not on the graph structure but on the "behavior" of node features, in particular their centroids.
The authors leverage geometric concepts to enhance class representation and model interpretability. Each class is represented through probabilistic centroids and geometric centroids organized within simplexes in lower-dimensional spaces. This approach offers new insights into why it is challenging to achieve significant results on certain datasets, identifying simple strategies that can improve node classification performance in GNN models, such as in the case of degenerate simplex and the feature re-shuffling phenomenon.

**Strengths:**

The first strength is that the paper introduces a new approach to studying datasets, which in turn allows for the analysis of GNN performance based on input datasets. I also find it particularly interesting that the method is entirely based on node features, making the study simple and fast.

I also believe that other strengths include the solid theoretical support provided by the authors and, above all, the practical aspect of the study, which proposes simple and effective strategies capable of improving GNN performance as well as model interpretability.

**Weaknesses:**

The paper is well-written and rich in content; however, it is somewhat difficult to follow due to the abundance of theoretical contributions and the numerous references to the appendix. I understand, however, that with the page limit, it is challenging to organize all the material effectively.

It should also be noted that some assumptions are rather optimistic: for instance, even in the simple Lemma 1, I would expect some overlapping in reality. Nonetheless, I understand that this is intended as a starting point.

Another shortcoming is that the studies conducted so far have limited applicability, as the approach mainly focuses on node features and datasets used for node classification.  I was wondering if it might be possible to gain some insight in the case of graph classification as well.

It also seems that a thorough discussion on scalability is lacking (or maybe I missed it): what happens, for instance, when the graphs are not very large or when the features are low-dimensional?
I did not notice a section dedicated to the actual limitations of the method.

**Questions:**

In Figure 5, I would add class centroid as well.
I would also use in fig. 1 Squirrel.

---

> ### Author Response · Authors · 2024-11-21
> **Response to reviewer's comments Part 1**
>
> Thank you for the comments and suggestions.
>
> **Comment 1:** The paper is well-written and rich in content; however, it is somewhat difficult to follow due to the abundance of theoretical contributions and the numerous references to the appendix. I understand, however, that with the page limit, it is challenging to organize all the material effectively.
>
> **Response:** Thank you for your comments. We agree with the reviewer that due to the page limit, it is impossible to include all content in the main body of the paper. Therefore, we have included most numerical evidence supporting our theory in the Appendices. We aim to dedicate the main body to theoretical discussions and explain the *intuitions* and *implications*.
>
> **Comment 2:** It should also be noted that some assumptions are rather optimistic: for instance, even in the simple Lemma 1, I would expect some overlapping in reality. Nonetheless, I understand that this is intended as a starting point.
>
> **Response:** We understand the reviewer's concern. We have verified that Lemma 1 holds for *all* the datasets in Sec. 5 Experiments, as well as standard homophilic datasets such as Cora, Citeseer, PubMed.
>
> The assumptions are not as unrealistic as one may expect, as we discuss in Appendix B. There are two categories of features that fit the assumptions: (1) If each component of a feature is either $0$ or $1$ (e.g., Cora, Citeseer, etc.), then features of nodes are on a high-dimensional hypercube. In this case, the features are always in a convex position. (2) If each component of a feature is randomly chosen from $(0,1)$ (here we omit a few technicalities), then it is known that with high probability, the features are in a convex position provided the feature dimension is sufficiently large. To get a sense of "sufficiently large", if the feature dimension is $30$, then $10^7$ features are in a convex position with probability $\geq 99\\%$ [a]. For most ML applications, feature dimensions are likely to be $>30$ (except for some special cases), which makes Lemma 1 widely applicable.
>
> We agree with the reviewer that our paper proposes a feature-centric theoretical framework to analyze GNNs. Therefore, assumptions on features are needed as a starting point. As we have discussed above, we believe they are not particularly stringent.
>
> [a]: A. Gorban, et. al. Stochastic separation theorems: How geometry may help to correct AI errors, Notice AMS, 70(1):25-33, 2022.
>
> **Comment 3:** Another shortcoming is that the studies conducted so far have limited applicability, as the approach mainly focuses on node features and datasets used for node classification. I was wondering if it might be possible to gain some insight in the case of graph classification as well.
>
> **Response:** We have pointed out in the paper that our focus is on node classification, and proposed tricks such as re-wiring and early stopping are irrelevant for graph classification. For re-wiring, it changes the graph structure (e.g., the molecular structure of chemicals), which is undesirable. The proposed early stopping is solely to tackle heterophily, which is not an issue of graph classification.
>
> However, the feature normalization trick (assuming there are meaningful node features, e.g., features of atoms in a molecule) may still help as it serves the same purpose of making the simplex $\Delta_g$ more regular, thus making the node features more distinguishable. We have tested on MUTAG, Proteins, and ENZYMES datasets with base models GIN [b] and PANDA [c]. We observe slight improvement over the base models, however, it also prompts exploring more principled approaches based on the feature-centric framework to get better results. We will discuss this as a limitation of our paper in the revision (see our response to Comment 4).
> | Method   | MUTAG           | Proteins        | ENZYMES         |
> |----------|------------------|-----------------|-----------------|
> | GIN      | 88.83±6.10      | 75.60±1.34      | 48.35±4.47      |
> | GIN-N    | 90.11±4.35      | 75.71±1.18      | 49.52±4.91      |
> | PANDA    | 88.49±4.90      | 74.46±2.22      | 46.33±4.73      |
> | PANDA-N  | 89.50±3.86      | 75.51±2.70      | 47.16±4.13      |
>
> [b]: K. Xu, W. Hu, J. Leskeovec and S. Jegelka. How powerful are graph neural networks? ICLR, 2019.
>
> [c]: J. Choi, S. Park, H. W, S. Cho and N. Park. PANDA: expanded width-Aware message passing beyond rewiring. ICML, 2024.

---

> ### Author Response · Authors · 2024-11-21
> **Response to reviewer's comments Part 2**
>
> **Comment 4:** It also seems that a thorough discussion on scalability is lacking (or maybe I missed it): what happens, for instance, when the graphs are not very large or when the features are low-dimensional? I did not notice a section dedicated to the actual limitations of the method.
>
> **Response:** The proposed framework is graph-agnostic and does not have any scalability issue w.r.t.\ graph size. The proposed tricks are simple and the scalability largely depends on the base model used. Our numerical studies have included both small graphs ($183$ nodes) and large networks ($>400000$ nodes and $>1.35$ million edges).
>
> Regarding feature size (see also Comment 2 above), the assumption of our framework may fail if the feature dimension is very small. This is the case for the "genius" dataset in the appendix. However, from the results in Table 11, the proposed tricks work reasonably well on "genius". (Notice there is a typo for "Penn94", the feature dimension is $4814$ instead of $5$.) To test the performance of the proposed tricks for small feature sizes, we have conducted the study by randomly removing a fraction of features. We choose the best base model ACM-GCN (see Table 2) and consider retaining $1/2, 1/4, 1/8$ feature dimensions. The results are shown in the table below and we see that the tricks generally improve the performance.
>
> **Table:** Node classification results (%).
>
> | Feature Dimension | Full           | 1/2           |1/4          | 1/8        |
> |-------------------|----------------|---------------|---------------|---------------|
> | **Chameleon**     |                |               |               |               |
> | ACM-GCN           | 71.29±8.06     | 61.95±7.13    | 52.70±5.36    | 44.17±3.87    |
> | ACM-GCN-AEN       | 75.22±9.99     | 63.53±7.37    | 56.09±5.36    | 46.07±4.92    |
> | **Squirrel**      |                |               |               |               |
> | ACM-GCN           | 55.07±8.96     | 49.67±7.43    | 42.10±5.08    | 36.27±2.72    |
> | ACM-GCN-AEN       | 59.47±11.41    | 50.49±7.81    | 44.10±5.93    | 37.67±2.38    |
> | **Actor**         |                |               |               |               |
> | ACM-GCN           | 37.28±2.76     | 33.74±2.34    | 31.32±1.67    | 26.89±0.97    |
> | ACM-GCN-AEN       | 47.62±6.38     | 35.09±1.91    | 32.33±1.53    | 29.23±0.85    |
>
> We will include the following short section on the limitations of our paper:
>
> "**Limitations:** The framework gives us a glimpse into theoretical explanations of many GNN phenomena and enhances our understanding of the mechanisms and fundamental limitations of GNN models. Given a dataset, we also aspire to use the framework to rigorously identify the hardness (quantitatively). From our theoretical and numerical results, the quasi-isometric shape of the feature centroid simplex demonstrates its usefulness in distinguishing easy datasets (e.g., Taxes) from hard ones (e.g., Actor). However, we are not able to prove that any numerical invariants associated with the shape can give a guaranteed error (lower) bound, i.e., the Bayesian error, for any GNN models. We will investigate further in future works.
>
> Technically, the reliance of our framework on node features faces challenges when there are few useful features. For example, this occurs for many graph classification tasks, where the label is solely determined by the graph structure. For such a task, our proposed framework does not immediately give an insightful understanding of the dataset. A possible solution is to apply the framework in conjunction with position encoding of the nodes as features. "
>
> **Comment 5:** In Figure 5, I would add class centroid as well. I would also use in fig. 1 Squirrel
>
> **Response:** We will follow your suggestion when revising the paper.

---

> ### Author Response · Authors · 2024-11-25
>
> Dear Reviewer kubS,
>
> As the discussion period nears its end, we hope that we have effectively addressed and resolved your concerns. Your feedback on our rebuttal responses would be greatly appreciated, and we are more than willing to provide further clarification on any remaining issues. Thank you for your time and consideration.

---

> > ### Comment · Reviewer_kubS · 2024-11-26
> >
> > I found the authors’ responses clear and satisfactory, so I raised the final score.

---

> > > ### Author Response · Authors · 2024-11-26
> > >
> > > We sincerely thank you for taking the time to review our revised manuscript and for your thoughtful feedback. Your constructive comments have significantly contributed to improving the quality of our work, and we deeply value your support.

---

### Author Response · Authors · 2024-11-22
**Paper revision**

We would like to thank all reviewers for their insightful comments and suggestions. In view of them, we have prepared and uploaded a revised version of our paper with the following major updates:
1. We have reorganized the paper, particularly Section 2 and Section 3. Our definitions of model geometric shapes are now introduced together at the end of Section 2. We have rewritten parts of Section 2.1, focusing on the intuitions rather than technical details (which can still be found in the Appendix).
2. We have included new numerical results at appropriate places of the revision based on the comments from all the reviewers.
3. We have included the new Appendix G to discuss the limitations of our work and expanded Appendix C on related works.

---

### Meta-Review · Area_Chair_SZaH · 2024-12-14

**Metareview:**

This submission assumes a novel perspective for graph neural networks, focusing on the *geometrical* aspects of node features of graphs (as opposed to a perspective driven by local topological information via message passing, for instance). This permits studying existing phenomena through a different lens, leading to new insights and explanations. Of particular relevance to machine-learning practitioners is a section on how to employ simple "tricks" for improving predictive performance in graph-learning tasks.

The primary strengths of the paper lie in...
1. ...building a new perspective built on node feature information,
2. ...phrasing existing results in the language of *coarse geometry*, and
3. ...an empirical validation and verification of the utility of new perspective.

The main weaknesses of the paper, some of which had already been identified by reviewers, involve...
1. ...a lack of "accessibility" given the strong theoretical perspective,
2. ...several strong assumptions in the theoretical part, and
3. ...a focus on node-classification experiments.

Concerns about these weaknesses could be largely alleviated during the rebuttal phase, and strongly agree with all reviewers that the paper is now ready for publication. The authors are to be commended for their willingness to engage with reviewers. I trust them to incorporate the promised changes, in particular those that concern **accessibility** and **contextualisation of the assumptions** in their revision. I believe that this will help make the work available to a larger community of readers, which shall ultimately be highly beneficial for the work as such.

**Additional Comments On Reviewer Discussion:**

The main weaknesses concerned (perceived) novelty (`NSvv`, `ZoMg`) and the contextualisation of the work (`NSvv`), accessibility (`kubS`, `ZoMg`, `p71D`), and theoretical assumptions (`kubS`).

The authors were able to provide sufficient feedback and revisions to alleviate the concerns by reviewers; the reviewers were in particular impressed by the textual improvements observed during the revision—I share this sentiment!

Following the discussion phase, I did not consider one point of reviewer `kubS`, viz. the focus on node classification task, to be too problematic (even though it is a valid criticism!). In fact, I would suggest the authors to keep the focus as-is (despite their interesting results shown during the rebuttal), mostly for the reason that I believe that a single-focus paper will be easier to read and understand, given the space limitations. Thus, overall, reaching my final verdict proved easy for this paper: It is clear that this is a timely, relevant work whose novel perspective will prove useful. As outlined above, I trust the authors to implement the promised changes and further work on the **presentation** of the paper, which I believe to be the most critical aspect at this point. Striving for a paper that reaches a broad swathe of researchers in graph learning will only benefit the work and its potential extension.

---

### Decision · Program_Chairs · 2025-01-22

Accept (Poster)